# Cotton plant defence against a fungal pathogen is enhanced by expanding BLADE-ON-PETIOLE1 expression beyond lateral-organ boundaries

Zhennan Zhang[1,4], Peng Wang[1,2,4], Xiaoli Luo[1,3,4], Chunlin Yang[1], Ye Tang[1], Zhian Wang[1,3], Guang Hu[1], Xiaoyang Ge[2], Guixian Xia[1] & Jiahe Wu [1]

In the plant response to pathogen infection, many genes' expression is temporally induced, while few spatially induced expression genes have been reported. Here, we show that GhBOP1 can autonomously expand expression from restrained tissue when *Gossypium hirsutum* plants are attacked by *Verticillium dahliae*, which is considered to be spatially induced expression. Loss- and gain-of-function analyses show that GhBOP1 is a positive regulator in the modulation of plant resistance to *V. dahliae*. Yeast two-hybrid assays, luciferase complementation imaging and GUS reporting show that GhBOP1 interaction with GhTGA3 promotes its activation activity, regulating the expression of down-stream defence-related genes. Moreover, the induced spatial expression of GhBOP1 is accompanied by GhBP1 repression. Both antagonistically regulate the lignin biosynthesis, conferring cotton plants enhanced resistance to *V. dahliae*. Taken together, these results demonstrate that GhBOP1 is an economic positive regulator participating in plant defence through both the GhBOP1-GhTGA3 module and lignin accumulation.

[1] The State Key Laboratory of Plant Genomics, Institute of Microbiology, Chinese Academy of Sciences, 100101 Beijing, China. [2] The State Key Laboratory of Cotton Biology, Institute of Cotton Research, Chinese Academy of Agricultural Sciences, 455000 Anyang, Henan, China. [3] Institute of Cotton Research, Shanxi Agricultural Academy of Sciences, 044000 Yuncheng, China. [4]These authors contributed equally: Zhennan Zhang, Peng Wang, Xiaoli Luo Correspondence and requests for materials should be addressed to G.X. (email: xiagx@im.ac.cn) or to J.W. (email: wujiahe@im.ac.cn)

Plants stand and face various biotic and abiotic stresses; therefore, they have evolved a wide spectrum of mechanisms to constantly defend themselves against these stresses, especially pathogen infestation. The mechanisms in response to pathogen attack are divided into two groups, pre-existing and induced. The pre-existing mechanisms mainly involve physical and chemical barriers for protecting plants from pathogen infestation at the first line, including the plant cuticle, cell wall, and antimicrobial compounds[1]. Salicylic acid (SA) crosstalking with auxin, ethylene, and jasmonates (JA) is crucial for the inducible response of plants to pathogens infection including systemic acquired resistance and induced systemic resistance[2–5]. In disease resistance induced mechanism, NON-EXPRESSOR OF PATHOGENESIS-RELATED GENES1 (NPR1) is regarded as a major knotting component to regulate plant defence, required for SA perception and other hormones[6,7]. NPR1 is a BTB-ankyrin protein including two conserved motifs: a BTB/POZ (for Broad Complex, Tramtrack, and Bric-a-brac/POX virus and Zinc finger) domain at the N-terminus and four ankyrin motifs near the C-terminus. These inducible responses of plant to pathogens attack involve in expression change of lots of genes in temporal mode. However, gene expression change in spatial mode remains to be less known.

BTB-ankyrin proteins are a small family in plants. In Arabidopsis, there are six BTB-ankyrin proteins[8]. NPR1, NPR3, and NPR4 through SA perception have a role in defence[7,9,10], and NPR2 has been described to have a secondary role in SA perception[11]. BLADE-ON-PETIOLE1 (BOP1) and BOP2 are the other two BTB-ankyrin proteins, the expression of which is generally restrained in lateral-organ boundaries (LOBs). BOP1/2 is described to have an important role in the development of the leaf and inflorescence architecture[12,13]. A meta-analysis of sequencing data showed that BTB-ankyrin proteins originated prior to the emergence of land plants[14]. All land plants sequenced genomes, including primitive mosses, encode homologs of both NPRs and BOP1/2, indicating that the ancestral BTB-ankyrin proteins may have held functions in both defence and development[15]. The initial characterization of the *bop1bop2* mutant showed no change in resistance to pathogens[16]. However, a recent report has indicated that BOP1/2 participates in plant defence through the JA signalling pathway[17]. Thus, the function of BOP1/2 in resistance to pathogens needs to be further evaluated.

BOP1/2 and NPR1 share homologous functional domains, BTB/POZ and ankyrin repeats, that potentially support a similar mode of action. Thus, the NPR1 signalling mechanism potentially serves as a paradigm for BOPs. BTB-ankyrin proteins including NPR1 and BOP1 can interact with defence-related TGA bZIP transcription factors to exert functions in disease resistance[16,17]. TGA bZIP proteins are a distinct subclade in the bZIP superfamily[18]. In Arabidopsis, these defence-related TGAs contain three clades: Class I comprises TGA1 and TGA4; Class II comprises TGA2, TGA5, and TGA6; and Class III comprises TGA3 and TGA7[19]. NPR1 interaction with TGAs participates in plant defence, which should represent a similar interaction mode for the function of BOPs in pathogen resistance. Weak physical interactions have been detected between BOPs and most TGAs in a yeast heterologous system[16,20,21]. A recent report has shown that the change of BOP interacts with TGAs in yeast indirectly caused by co-expression of npr1-1[14,17]. Subsequently, the methyl jasmonate-induced resistance is abolished in *bop1bop2* mutants and enhanced in plants overexpressing BOP1 or BOP2, possibly through the ability of NPR1 to disrupt BOP interactions with TGAs[14,17]. Thereby, the BOP-TGA interaction is considered to have confirmed roles in defence and development.

Verticillium wilt is a highly destructive vascular disease caused by *Verticillium* sp, a soil-borne fungus, which infects a wide range of plants[22,23]. Mounting evidence has confirmed that verticillium wilt resistance is directly associated with lignin accumulation in plants. In cotton, the series of lignin synthesis enzymes are upregulated when the plants are infected by *V. dahliae*, resulting in lignin accumulation[22,24–27]. For example, transcript levels of lignin metabolism-related genes are increased in cotton plants after inoculation with *V. dahliae*[24]. GbERF1-like regulates lignin metabolism-related gene expression for lignin accumulation, increasing the resistance to *V. dahliae* infection[26]. In Arabidopsis, plants inoculated with *V. longisporum* promote novel vascular formation and lignin synthesis[28]. The Ve-mediated resistance response of tomato to *V. dahliae* also involves lignin and PAL gene expression[29]. These reports document that lignin plays important roles in plant defence. Interestingly, it has been reported that BOPs participate in lignin synthesis by regulating lignin metabolism-related gene expression to show phenotypes in *bops* mutants and overexpression plants[13]. However, it remains unclear whether BOPs participate in disease resistance through lignin synthesis.

BOPs function in lignin synthesis in association with some gene regulation, including BP1 in Arabidopsis[13,16]. When *BP1* expression is repressed in stem, *BOP1/2* exhibits ectopic expression outside of LOBs, suggesting that BOPs and BP1 have antagonistic roles in regulating lignin biosynthesis[13,30]. Seven lignin biosynthesis genes with upregulated expression in *bp1* mutants (*PAL1*, *C4H1*, *4CL1*, *C3H1*, *CCoMT1*, *CAD5*, and *PRXR9*) can be restored to near wild-type levels by *bop1bop2* mutation; similar to this result, in BOP overexpression plants, five of the seven genes are dramatically upregulated, which is consistent with the promotion of lignin biosynthesis[13,30]. Therefore, BOPs and BP1 together regulate lignin synthesis, potentially participating in plant disease resistance.

In the present study, we found that cotton *BOP1* expression was upregulated and autonomously expanded outside of LOBs coupled with *BP1* repression during *V. dahliae* infestation. The results of genetic and biochemistry experiments showed that GhBOP1 was an economical regulator participating in plant defence against *V. dahliae*, resulting from the interaction with GhTGA3 and lignin accumulation. Taken together, these data shed light on the molecular mechanisms of BOPs function in plant defence, including BOP1-TGA3, similar to the NPR1-TGA module and antagonistic roles of BOP1 and BP1 in lignin synthesis.

## Results

**GhBOP1 autonomously expanding expression infected by fungus.** In cotton RNA-sequencing of the plant response to *V. dahliae* infection, the *NPR1-like* (BOPs) expression level was significantly induced[31], and BOPs possessed the same domains and a similar structure as NPR1 (Supplementary Fig. 1a), which led us to dissect the function of BOPs in cotton plant response to pathogen infection. In *G. hirsutum*, there are two homologues of BOP proteins, GhBOP1 and GhBOP2. *GhBOP1* has two copies, Gh_A09G1115 and Gh_D09G1120 located in At and Dt subgenomes, respectively, sharing 99.6% and 99.4% similarities in coding sequences and amino acid sequences, respectively (Supplementary Figs. 2 and 3). Thus, Gh_A09G1115 were chosen for researching characterization of GhBOP1. GhBOP1 and GhBOP2 (Gh_A01G1644) are highly conserved with other plant BOPs (Supplementary Fig. 1b). GhBOP1/2 proteins have a strong relationship with two BOPs in *Populus Trichocarpa* belonging to BTB-ankyrin protein family, as well as NPRs (Fig. 1a). *GhBOP1* expression in roots was significantly induced after *V. dahliae* inoculation, while the *GhBOP2* expression level was moderately increased (Fig. 1b). The results of the virus-induced gene

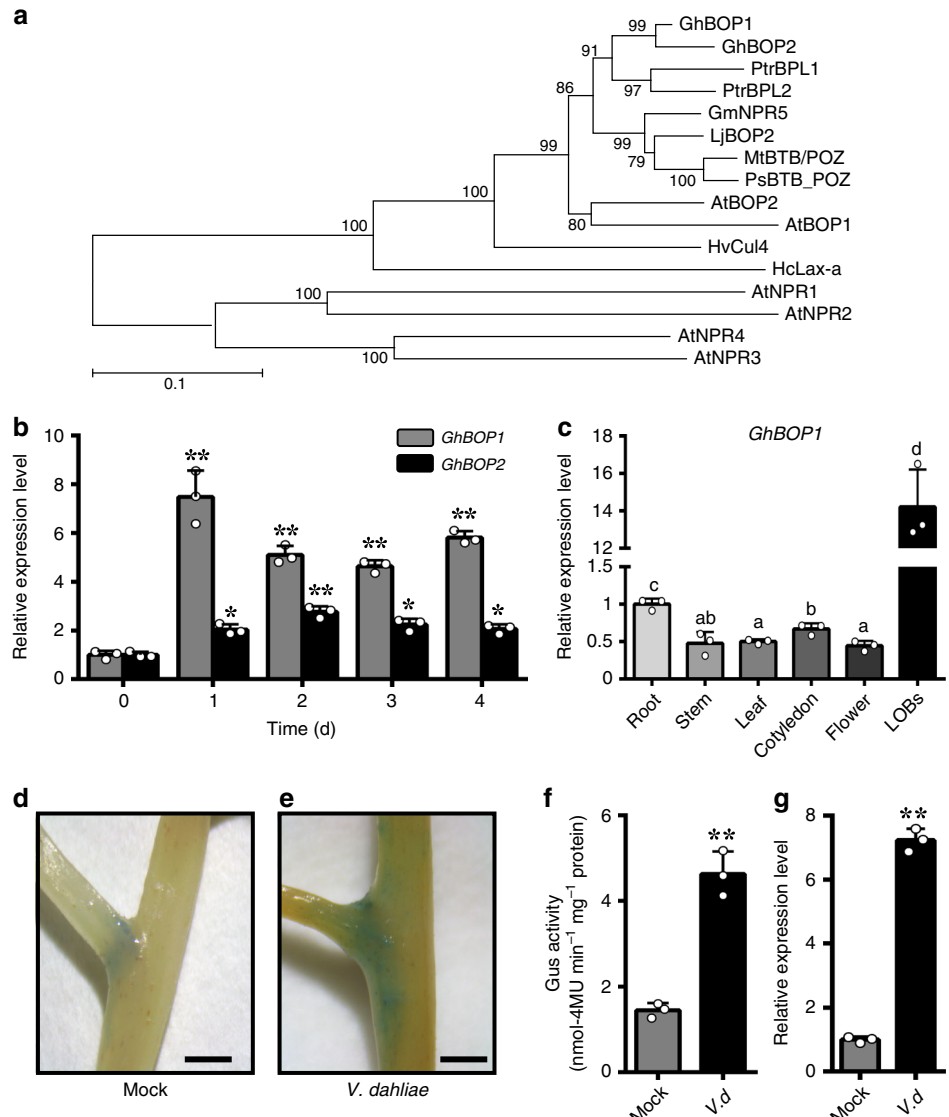

**Fig. 1** Analysis of *GhBOP1* specific expression in the lateral-organ boundaries and autonomously expanding expression in response to *V. dahliae* infection. **a** Phylogenetic tree of BOP proteins from *G. hirsutum* and other species. The complete amino acid sequences of BOPs were aligned using ClustalX and assessed with MEGA 5.0 using the neighbour-joining method with 1000 bootstrap replicates. The numbers next to each node represent confidence percentages. Branch lengths are proportional to the amount of inferred evolutionary change. **b** Expression patterns of *GhBOP1* and *GhBOP2* induced by *V. dahliae*. Total RNA was extracted from roots at 0, 1, 2, 3, and 4 day post-inoculation. *GhUB7* served as an internal control. Error bars represent the SD of three biological replicates. Asterisks indicate statistically significant differences compared to respective 0 d, as determined by Student's *t*-test (*$P < 0.05$, **$P < 0.01$). **c** Transcript levels of *GhBOP1* in different tissues of cotton. Total RNA was extracted from root, stem, leaf, cotyledon, flower, and LOBs. *GhUB7* served as an internal control. Error bars represent the SD of three biological replicates. The different letters indicate statistically different means at $P < 0.05$ (one-way ANOVA with a Duncan post-hoc test). **d**, **e**, **f** GUS staining of the LOBs of GhBOP1pro:GUS transgenic plants at 0 day (**d**) and 3 day (**e**) after *V. dahliae* inoculation, and GUS activity analysis (**f**), respectively. The scale bars indicate 1 mm. Error bars represent the SD ($n = 12$) of three biological replicates. Double asterisks indicate statistically significant differences, as determined by the Student's *t*-test (**$P < 0.01$). **g** *GhBOP1* expression analysis of the LOBs of seedlings at 3 day after *V. dahliae* inoculation. Error bars represent the SD of three biological replicates. Statistical analysis was performed using the Student's *t*-test (**$P < 0.01$)

silencing (VIGS) analysis showed that *GhBOP1*-silenced plants had an increased sensitivity to *V. dahliae* infection, while *GhBOP2*-silenced plants were similar to the control with respect to disease symptoms (Supplementary Fig. 4). Thereby, GhBOP1 was considered to potentially participate in plant defence against *V. dahliae*.

In Arabidopsis, previous reports have shown that *BOP* gene expression is restrained in LOBs[14,16,20,32–34]. Using qPCR analysis, *GhBOP1* was dominantly expressed at the LOBs compared with other tissues (Fig. 1c). Analysis of a GhBOP1pro:

GUS reporter gene showed that the GUS staining spots were mainly focused at the boundary between the stem and brunch, suggesting that *GhBOP1* expression was restrained in LOBs (Fig. 1d).

Transgenic plants carrying the GhBOP1pro:GUS reporter gene were also analysed in response to fungal inoculation. The GUS staining colour clearly deepened at the boundary between the stem and branch of plants inoculated with *V. dahliae* compared with the mock control, which was consolidated by the GUS quantitative analysis (Fig. 1e and f). More interestingly, GUS

staining spots expanded beyond the LOBs by *V. dahliae* induction (Fig. 1e and f). Compatible with the GUS staining analysis, the *GhBOP1* expression level in the zones around the boundary was 7.5-fold higher than in the mock control (Fig. 1g). Additionally, the GUS staining color in root after inoculated with *V. dahliae* was darker than that of the mock treatment (Supplementary Fig. 5), indicating the GhBOP1 expression level in roots was induced by the fungi. These data suggested that the expression of the *GhBOP1* gene was spatially induced by *V. dahliae*.

**GhBOP1 positively regulates plant defence against *V. dahliae*.** To evaluate the function of GhBOP1 in plant response to pathogen, *GhBOP1* RNAi and gain-of-function plants were generalized using *A. tumefaciens*-mediated transformation methods. Two out of 24 independent *GhBOP1* RNAi transformants, bop1-8 and bop1-13, showed lower *GhBOP1* transcriptional levels, (Supplementary Fig. 6a). The bop1-8 and bop1-13 contained one T-DNA copy in the genome by southern blot identification (Supplementary Fig. 6b). These adult RNAi plants were slightly slender in architecture compared with the wild-type (WT) plants (Supplementary Fig. 6c). The knockdown seedlings were more sensitive to *V. dahliae* infection, with more severe defoliation and yellowing symptoms compared with the WT plants (Fig. 2a). The RNAi plants displayed a larger ratio of disease grade 3 and 4 than the WT plants (Fig. 2b). The brown spots in the vascular tissue of the RNAi stem slope sections were more intense compared to the WT (Fig. 2c). Additionally, the *GhBOP1*-knockdown plants had a greater fungal biomass than in the WT (Fig. 2d). The fungal recovery assay showed that the stem sections from RNAi plants inoculated with *V. dahliae* had more fungal growth than the inoculated WT (Fig. 2e). These data indicated that GhBOP1 positively regulated plant resistance to *V. dahliae*.

To further investigate GhBOP1 function, two out of 26 independent *GhBOP1* overexpression transformants, OB1-1 and OB1-5, with higher *GhBOP1* transcriptional levels and one T-DNA copy insertion in the genome were selected (Supplementary Fig. 6a and b). The *GhBOP1*-overexpressing adult plants showed a slightly shorter plant height than the WT (Supplementary Fig. 6c). The GhBOP1 gain-of-function plants inoculated with *V. dahliae* exhibited an obvious resistance phenotype with less severe defoliation and yellowing symptoms compared with WT (Fig. 2a). Compared to WT, OB1-1 and OB1-5 showed lower proportion of disease grade 3 and 4, less brown colour intensity in the vascular tissue, less extent of fungal recovery and lower fungal biomass (Fig. 2a–e). Collectively, these results show that GhBOP1 is a positive regulator in the plant response to *V. dahliae* infestation.

We then analysed the expression levels of genes associated with the SA- or JA-mediated defence pathways in transgenic roots at 3 day after *V. dahliae* inoculation. As shown in Fig. 2f, the expression levels of the SA-mediated genes *GhPR1* and *GhPR2* were dramatically decreased in *GhBOP1*-RNAi plants compared with WT. Interestingly, the expression levels of the JA-mediated genes *GhPR3* and *GhPDF1.2* were also extremely reduced in knockdown plants. However, in overexpression plants, the expression levels of SA- and JA-mediated genes were clearly increased. These results indicate that GhBOP1 possibly participates in both SA- and JA-mediated defence pathways in cotton.

**Interaction of GhBOP1 with GhTGA3.** Given that BOP1 protein is similar to NPR1 in domain and structure, we considered the GhBOP1 defence function to be directly related its defence interaction partners in the TGA family, including TGA1 and TGA4 (subgroups I), TGA2 and TGA5 (II), and TGA3 (III). Therefore, the yeast two-hybrid was employed to examine the GhBOP1 interaction with GhTGA1, GhTGA2, GhTGA3,

GhTGA4, or GhTGA5. As shown in Supplementary Fig. 7, self-activation of GhTGAs was not observed. The yeast cells co-transfected with vectors harbouring GhBOP1-BD/AD and the negative control (Lam) did not grow on medium containing SD/–Ade/–His/–Leu/–Trp, while the yeast cells co-transfected with the positive control (p53) exhibited normal growth. The yeast cells with GhBOP1-BD/GhTGA3-AD grew normally on medium containing SD/–Ade/–His/–Leu/–Trp, whereas the diluted $10^{-3}$ yeast cells with GhBOP1-BD/GhTGA1-AD, GhBOP1-BD/GhTGA2-AD, GhBOP1-BD/GhTGA4-AD, or GhBOP1-BD/GhTGA5-AD hardly grew (Fig. 3a).

To consolidate the interaction of GhBOP1 with GhTGA3, a luciferase (Luc) complementation imaging assay was performed. As expected, the tobacco leaves co-injected with of NLuc-GhBOP1/CLuc-GhTGA3 exhibited strong fluorescence intensity (Fig. 3b). However, co-injection of NLuc-GhBOP1/CLuc-GhTGA1, NLuc-GhBOP1/CLuc-GhTGA4, NLuc-GhBOP1/CLuc-GhTGA2, or NLuc-GhBOP1/CLuc-GhTGA5 exhibited poor fluorescence, indicating that GhBOP1 might not interact with them or had a weak interaction. The same results were obtained for the quantity of fluorescence intensity of luciferase (Luc) complementation imagines calculated by pixel, as shown in Fig. 3c. These results show that GhBOP1 can strongly interact with GhTGA3 in cells.

**Dependence of GhBOP1 cellular distribution on GhTGA3.** Given the interactions between GhBOP1-GhTGA3 and considering the elements of the NPR1-TGAs mechanism as a framework in Arabidopsis[35,36], we sought evidence in support of GhBOP1-GhTGA3 in participating in plant defence. We first examined the subcellular localization of GhBOP1 and GhTGA3 in *Nicotiana benthamiana* epidemic cells. As shown in Fig. 4a, green fluorescence was retained in the nuclei of cells transiently expressing GhTGA3-GFP fusion protein, consistent with the subcellular localization of TGAs in Arabidopsis, while the GhBOP1-GFP fusion protein was found in both the cytosol and the nucleus, which is linked to Arabidopsis BOP1/2 localization[16].

To elucidate whether GhBOP1 localization was affected by GhTGA3 interactions, immune blot analysis was employed to assess the levels of the two proteins in the nucleus and/or cytoplasm. As shown in Fig. 4b, GhBOP1 proteins were present in both the nucleus and cytoplasm of cotton root cells, consistent with the subcellular localization of the GhBOP1-GFP fusion. However, in *GhTGA3*-silenced plants, the amount of GhBOP1 proteins in the nucleus was remarkably lower than in WT plants, whereas the protein content in the cytoplasm was comparable. To validate the dependence of the GhBOP1 nuclear distribution on GhTGA3, we also included a parallel experiment in which the plants were infected by the pathogen. After challenging with *V. dahliae*, the GhBOP1 protein contents in both the nucleus and cytoplasm of root cells increased compared with the unchallenged plants due to pathogen induction of GhBOP1 expression. In *GhTAG3*-silenced plants, the abundance of GhBOP1 proteins in the nucleus was much lower than in the control. These results indicate that the distribution or accumulation of GhBOP1 proteins in the nucleus is dependent on the presence of GhTGA3 proteins.

**GhBOP1 promotes GhTGA3 activation activity.** To check the DNA-binding activity of TGA3 and whether this binding could be enhanced by GhBOP1, electrophoretic mobility shift assay (EMSA) experiments were conducted with the recombinant protein TGA3 from *Escherichia coli* and the synthetic probe with the two core sequences TGACG (hereafter called the TGACG element)[37]. The EMSA experimental results showed that the

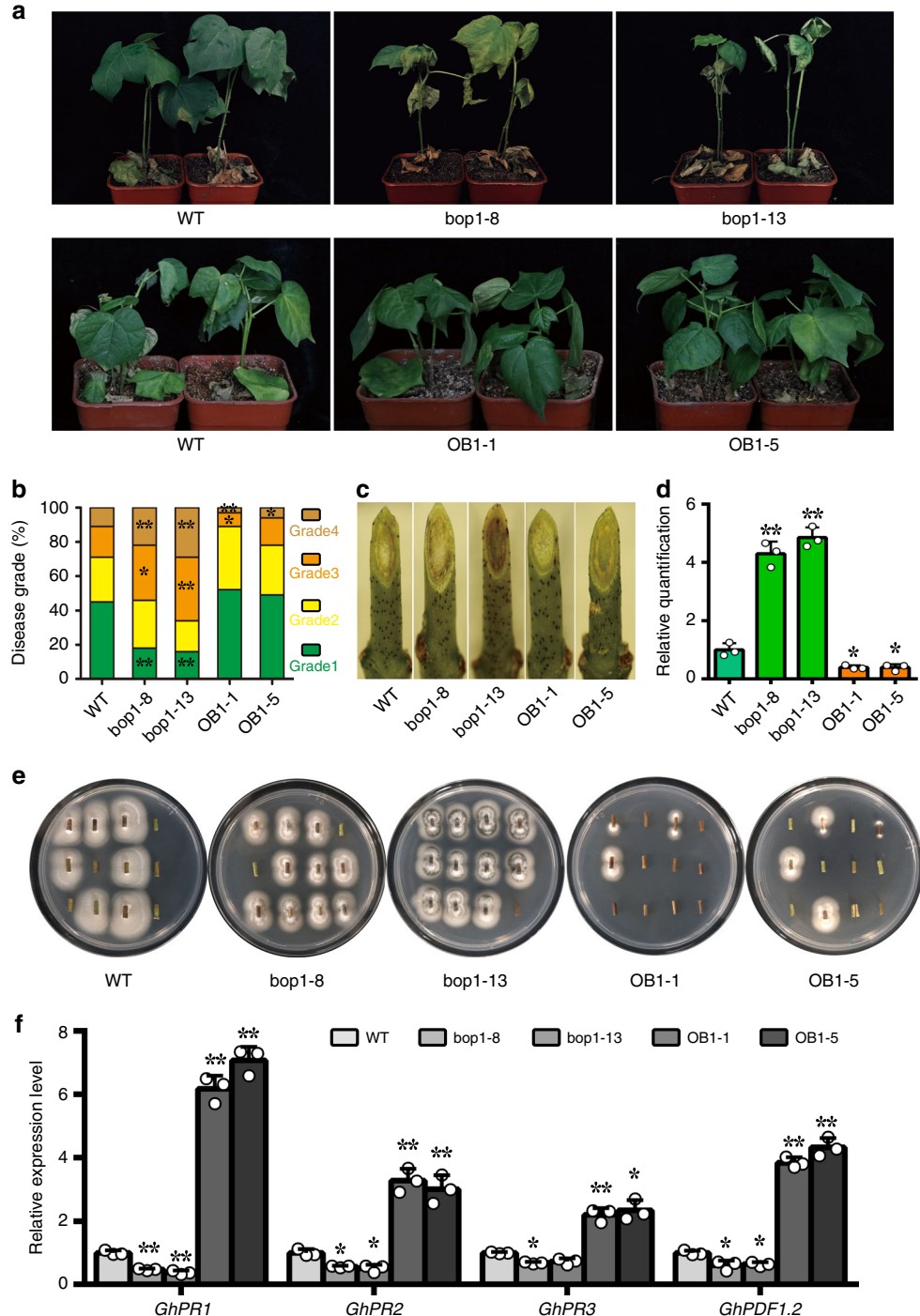

**Fig. 2** GhBOP1 positively regulates plant defence against *V. dahlia*. **a** The disease symptoms of WT, *GhBOP1*-RNAi, and -OE plants inoculated with *V. dahliae*. Images were obtained at 18 day after pathogen inoculation. **b** Disease grade analysis of the infected plants at 18 day after pathogen inoculation. The asterisks indicate statistically significant differences compared to corresponding disease grade of WT, as determined by the Student's *t*-test (\**P* < 0.05, \*\**P* < 0.01). **c** The oblique sections of stems revealed the disease symptoms in the vascular tissue of WT, *GhBOP1*-RNAi, and -OE plants. **d** Relative quantification of the fungal biomass in infected stems. qPCR analysis was conducted to compare the DNA contents between the *ITS* gene (measure of the fungal biomass) of *V. dahliae* and the *UB-7* gene of cotton (for equilibration) at 18 day post-inoculation. Error bars represent the SD of the three biological replicates. The asterisks indicate statistically significant differences, as determined by the Student's *t*-test (\**P* < 0.05, \*\**P* < 0.01). **e** Fungal recovery assay. The stem segments of inoculated plants were placed on PDA medium, and photographs were obtained at 4 day after culture. **f** Relative expression analysis of four resistance-related genes in WT and transgenic plants after *V. dahliae* inoculation. *GhUB7* served as an internal control. Error bars represent the SD of three biological replicates. The asterisks indicate statistically significant differences compared to corresponding gene expression level in WT, as determined by the Student's *t*-test (\**P* < 0.05, \*\**P* < 0.01)

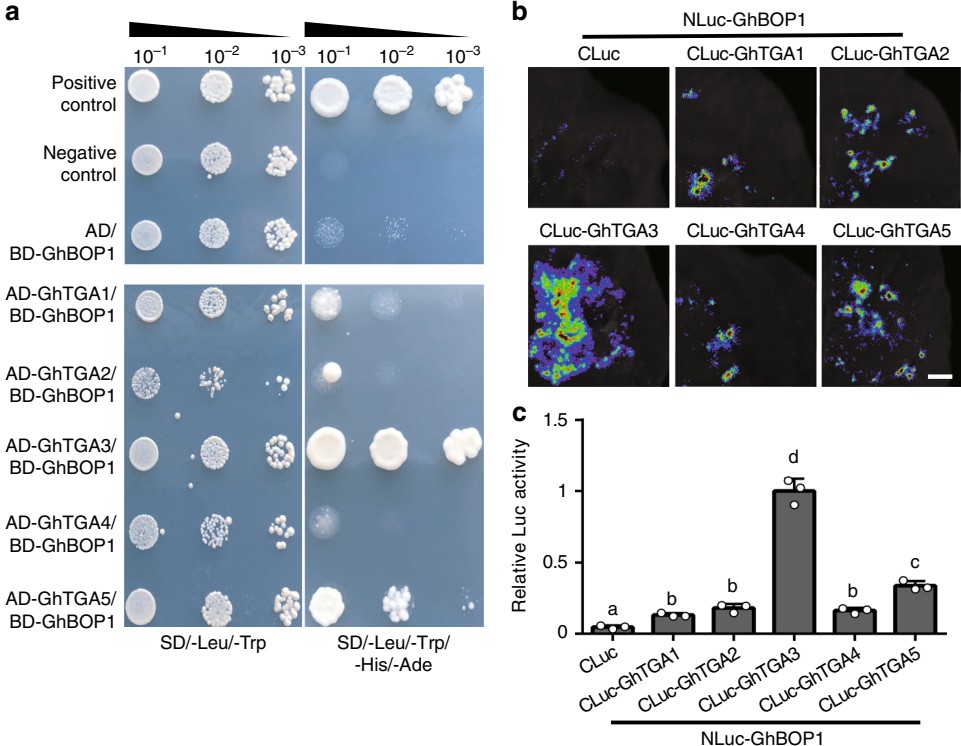

**Fig. 3** Interaction of GhBOP1 with GhTGA3. **a** Yeast two-hybrid assay to detect the interaction between GhBOP1 and GhTGA1/2/3/4/5. The yeast strains containing two correct plasmids were grown on SD/−Leu/−Trp plates and SD/−Leu/−Trp/−His/−Ade plates for 4 day. **b** Luciferase complementation imaging analysis of the interaction of GhBOP1 with GhTGA1/2/3/4/5. Agrobacterium strains containing the indicated vector were co-expressed in *N. benthamiana*. The luminescent imagines were taken at 36 h after infiltration. The scale bars indicate 2 mm. **c** Relative Luc activities in tobacco leaves measured with luminescence intensity by IndiGo software. In present NLuc-GhBOP1, the Luc activity of the CLuc-GhTGA3 in tobacco leaves was set to 1. Error bars represent the SD ($n = 12$) of three biological replicates. The different letters indicate statistically different means at $P < 0.05$ (one-way ANOVA with a Duncan post-hoc test)

retarded band of GhTGA3 protein binding the labelled probe was present, and the signal was gradually decreased by the addition of increasing amounts of unlabelled probe (Fig. 5a), demonstrating that GhTGA3 could bind specifically to the TGACG element in vitro.

To assess whether the interaction of GhBOP1 with GhTGA3 may have an effect on its binding activity, the binding analysis of GhTGA3 to the TGACG element was performed in the presence of recombinant GhBOP1 from *E. coli*. As shown in Fig. 5b, GhBOP1 alone did not bind to the probe, while an up-shifted band appeared when GhBOP1 was present in the reaction. Only the up-shifted band could be viewed when GhBOP1 was added up to 20 pM. These results further confirm that GhBOP1 can interact with GhTGA3, potentially enhancing transcription factor TGACG-binding activity in vitro.

To examine the transcription factor activity of GhTGA3, a dual-luciferase reporter assay system in Arabidopsis protoplasts was performed[38,39]. As shown in Fig. 5c, the relative value of GhTGA3 activating reporter expression was significantly higher than the control, demonstrating that GhTGA3 possesses high activation ability in vivo.

**GhBOP1 promotes GhTGA3 activation activity for *GhPR1*.** It is known that PR1 expression can be regulated by TGAs through LS5 and LS7 (containing the TGACG element) of the promoter[40]. In cotton, Gh_D09G0971 was identified as *GhPR1* based on the CottonGen database (https://www.cottongen.org/), the promoter of *GhPR1* (Gh_D09G0971), which contains a typical TGACG element at 846 bp upstream of the start codon according to the Plantcare (http://bioinformatics.psb.ugent.be/webtools/plantcare/

html/; Supplementary Table 1). In the loss- and gain-of-function of GhBOP1 plants, *GhPR1* transcriptional levels showed striking changes (Fig. 2f). Thus, we chose this gene as a target to assess the roles of GhTGA3 and GhBOP1 interactions using a transient expression assay in vivo. A promoter sequence of *GhPR1* replaced the CaMV35S promoter of vector pBIN121 to drive the GUS reporter gene. The Agrobacterium cells harbouring the indicated plasmids (Fig. 6a) were simultaneously injected into tobacco leaves. Two days later, the expression of *GhBOP1* and *GhTGA3* was confirmed by RT-PCR (Supplementary Fig. 8), and GUS expression was examined. As shown in Fig. 6a and b, the GhPR1 promoter itself drove GUS expression weakly, and the GUS staining intensity was ~3-fold higher when co-transformed with 35 S:GhTGA3. The staining intensity increased when 35 S:GhBOP1 was co-transformed with GhPR1pro:GUS, likely caused by the presence of tobacco TGA3 homolog(s), which might act in concert with GhBOP1 and stimulate GhPR1 promoter activity. When cells containing 35 S:GhTGA3, 35 S:GhBOP1, and GhPR1pro:GUS were simultaneously injected, the GUS staining intensity was much higher and reached an ~2-fold higher level than that in the cells co-transformed with GhPR1pro:GUS and 35 S:GhTGA3. These results demonstrate that GhTGA3 can activate *GhPR1* promoter activity, and GhBOP1 can in turn enhance this activity of GhTGA3 in vivo. To evaluate that the expression of *GhPR1* might rely on the functions of GhBOP1 and GhTGA3, GUS expression driven by the *GhPR1* promoter was tested in cotyledons of *GhBOP1*-RNAi and -overexpressing plants through transient expression. The protein in cotyledons agro-infiltrated with GhPR1pro:GUS was isolated, and GUS enzyme activity was monitored using two substrates, x-Gluc and 4-MUG.

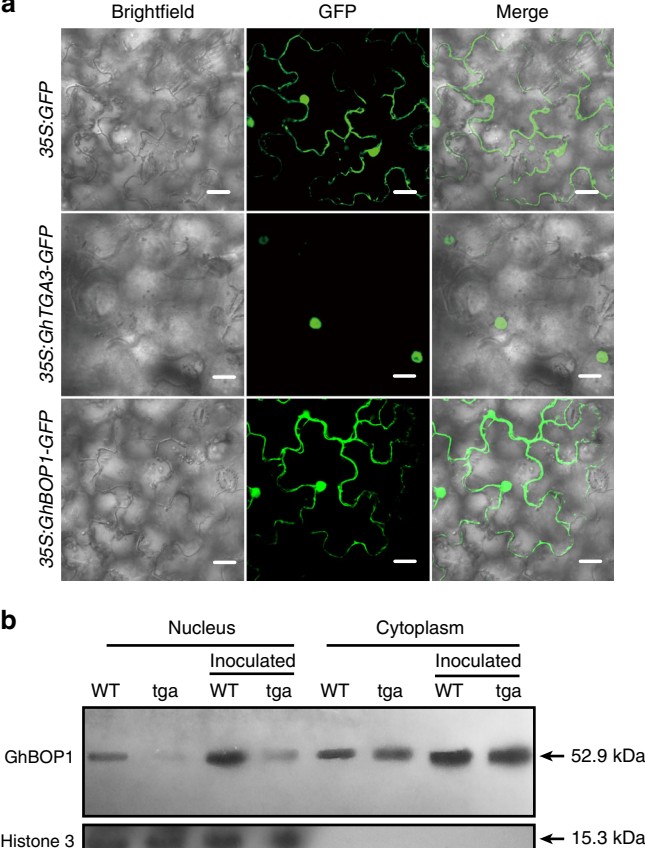

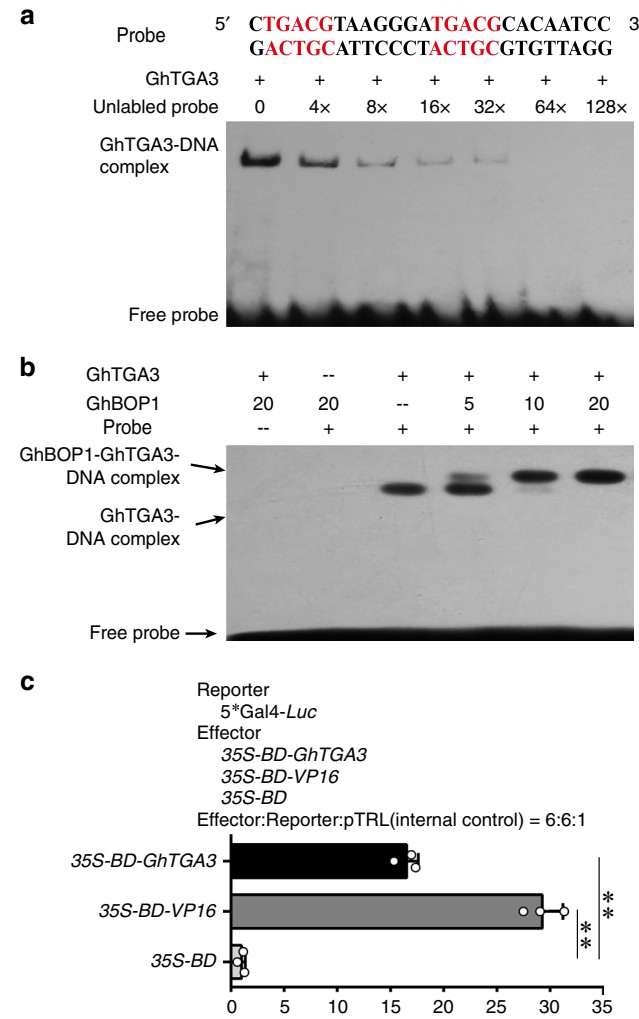

**Fig. 4** GhBOP1 cellular distribution possibly dependent on GhTGA3.
**a** Subcellular localization of GhGBOP1 and GhTGA3 in *N. benthamiana*. The scale bars indicate 20 μm. **b** The distribution of GhBOP1 between the nucleus and cytoplasm depended on the presence of GhTGA3. The nuclear and cytoplasm proteins were extracted from the roots of WT or *GhTGA3*-silenced plants after pathogen infection or mock treatment. Histone 3, the nuclear protein marker. β-Actin, the cytoplasm protein marker

**Fig. 5** GhTGA3 activation activity and GhBOP1 promotion potential.
**a** EMSA analysis of the specific binding of GhTGA3 to the probe containing the TGACG motif. GhBOP1 proteins were incubated with the biotin-labelled DNA probe in the reaction mixture for 30 min. The indicated amounts of unlabelled probe were used in the competition assay. The TGACG motif sequences are highlighted in red. **b** EMSA analysis of the efficiency of GhBOP1 in the binding activity of GhTGA3 to the TGACG motif. GhTGA3 proteins were incubated with the biotin-labelled probe, which contain two TGACG motifs in the present of GhBOP1. **c** Dual-luciferase reporter assay of the activation of GhTGA3 in Arabidopsis protoplasts. The effector vectors pRT-BD and pRT-BD-VP16 were used as negative and positive controls, respectively. Error bars represent the SD of three biological replicates. Each sample consisted of three technical repeats. Double asterisks indicate statistically significant differences, as determined by the Student's *t*-test (**$P < 0.01$)

As shown in Fig. 6c, the blue colour in the bop1-8 and bop1-13 reaction solutions with x-Gluc was lighter than in WT, while it was deeper in OB1-1 and OB1-5. When the *GhTGA3*-RNAi vector was co-injected in cotyledons with the GhPR1pro:GUS vector, the GUS enzyme activities of *GhBOP1*-overexpressing plants and WT significantly decreased compared with the cotyledon injected exclusively with GhPR1pro:GUS vector, while the enzymatic activity of *GhBOP1*-RNAi plants was comparable to those injected only with GhPR1pro:GUS vector (Fig. 6c). The relative blue intensity calculated by the absorbance at OD450 confirmed that GhBOP1 promoted the expression of downstream genes regulated by GhTGA3 (Fig. 6c). Consistent with these data, the results of the 4-MUG analyses also confirmed that GhBOP1 could enhance GhTGA3 transcriptional activity (Fig. 6d). These data support that both GhBOP1 and GhTGA3 function in the transcriptional regulation of downstream genes, including GhPR1, in cotton cells, implying that the transcriptional activation activity of GhTGA3 is coupled to the function of GhBOP1.

To further evaluate whether the positive regulation by GhBOP1 involving GhTGA3 in plant defence, the *GhTGA3*-silenced plants were produced by the VIGS method and challenged by *V. dahliae*. As shown in Supplementary Fig. 9, the *GhTGA3*-silenced plants exhibited increased sensitivity to the pathogens compared with the WT plants, showing serious disease symptom that included a

higher disease rate, more fungal recovery potential and a larger fungal biomass.

**GhBOP1 is a positive regulator of lignin synthesis**. In Arabidopsis, BOP1/2 promotes the expression of lignin biosynthetic genes, while, BP1 represses the expression of these lignin genes[41]. Recently, mounting evidence has shown that lignin accumulation confers an increased cotton plant resistance to *V. dahliae*[22,24–26,42]. Thus, it will be interesting to confirm biochemically and genetically whether GhBOP1 promotes the expression of lignin biosynthetic

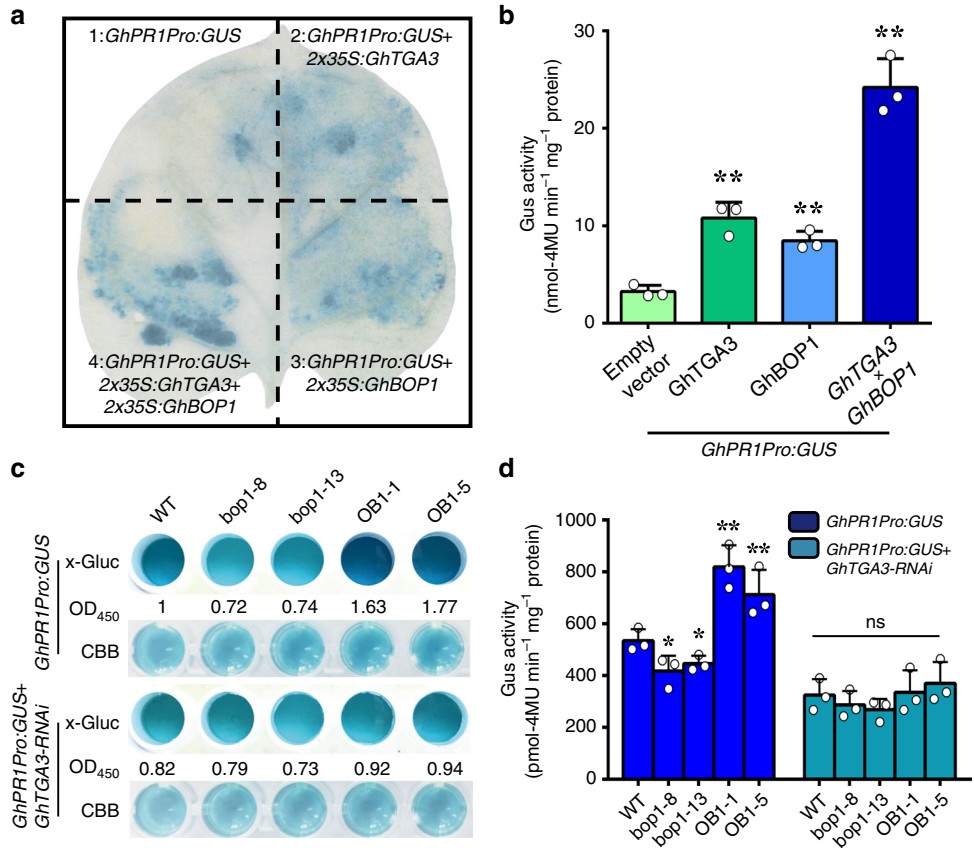

**Fig. 6** GhBOP1 enhances GhTGA3 activating activity in plants. **a** GUS staining analysis of GhBOP1 enhances GhTGA3 transcriptional activation to the GhPR1 promoter in tobacco leaf at 48 h after Agrobacterium infiltration with the indicated vectors. **b** Quantitative analysis of GUS activities in **a**. Error bars represent the SD ($n = 18$) of three biological replicates. Asterisks indicate statistically significant differences, as determined by Student's $t$-test (**$P < 0.01$). **c** GUS staining analysis of GhBOP1 enhancing GhTGA3 transcriptional activation to the GhPR1 promoter in WT and transgenic cotton cotyledon at 48 h after Agrobacterium infiltration with GhPR1pro:GUS or GhPR1pro:GUS and GhTGA3-RNAi, respectively. CBB (Coomassie brilliant blue) was used to normalize the protein extracted from tobacco leaves. The values indicate the relative blue densities of corresponding wells tested by OD450, the blue density of WT was normalized as 1. **d** GUS activity analysis in **c**. Error bars represent the SD ($n = 18$) of three biological replicates. Asterisks indicate statistically significant differences compared to WT, as determined by Student's $t$-test (*$P < 0.05$, **$P < 0.01$)

genes to participate in cotton plant resistance. As shown in Fig. 7a, the lignin biosynthetic genes, *PAL1*, *C4H1*, *4CL1*, *C3H1*, *CCoMT1*, and *CAD5*, showed significantly upregulation expression in over-expression plants compared with WT stems. In contrast, in RNAi plants, all these lignin biosynthesis genes showed downregulated expression, similar to previously reported results for plants inoculated with *V. dahliae*[22,24,25,27,42].

To further investigate the function of GhBOP1 in lignin accumulation, we stained cotton stems with phloroglucinol-HCl. Compared with the wild-type stems, the *GhBOP1*-overexpressing stems displayed more intense red staining (darker red) and a larger staining area, suggesting the presence of a relatively higher amount of lignin. In contrast, the RNAi plant stem sections showed the faintest red staining (pink) and a smaller staining area (Fig. 7b). To analyse the content of lignin in plant stems, we employed both the Klason and thioglycolate methods to measure the lignin content. As shown in Fig. 7c, we clearly observed different amounts of insoluble lignin (Klason lignin) from WT and transgenic stems. The mean dry weight of Klason lignin was 80.1 mg/g for wild type, 117.6 and 108.8 mg/g for OB1-1 and OB1-5, and 62.7 and 68.6 mg/g for bop1-8 and bop1-13, respectively (Fig. 7d). The results of lignin contents from the thioglycolate method were similar to results of Klason lignin (Fig. 7e).

Additionally, we tested the lignin accumulation of *GhBP1*-silenced plants, which provided similar results to *GhBOP1*-overexpressing plants with the two methods, as shown in Supplementary Fig. 10. These results document that GhBOP1 is a positive regulator in lignin deposition and that its accumulation may be involved in the negative regulation of GhBP1, consistent with the Arabidopsis lignin deposition[13]. Thus, the expanded GhBOP1 expression out of LOBs confers enhanced plant resistance to *V. dahliae*, partially due to the promotion of both lignin accumulation as well as GhTGA3 transcriptional activity.

**Expanding *GhBOP1* expression out of LOB by *GhBP1* repression.** The BOP-induced spatial expression during fungal infestation was similar to the phenotype observed in Arabidopsis bp1 mutants, reminiscent of the modulation of *BP1* repression[13]. Thus, we tested the expression levels of the cotton *BP1* gene in plants inoculated with *V. dahliae*. The *GhBP1* expression levels were significantly decreased post-inoculation compared with the control (Fig. 8a). These results indicated that *GhBP1* expression was extremely repressed during fungus infection, which possibly resulted in ectopic *GhBOP1* expression.

To further characterize the function of GhBP1 in the expanded expression of GhBOP1, the expression profiles in *GhBP1*-silenced plants were tested. The *GhBP1* expression level was significantly

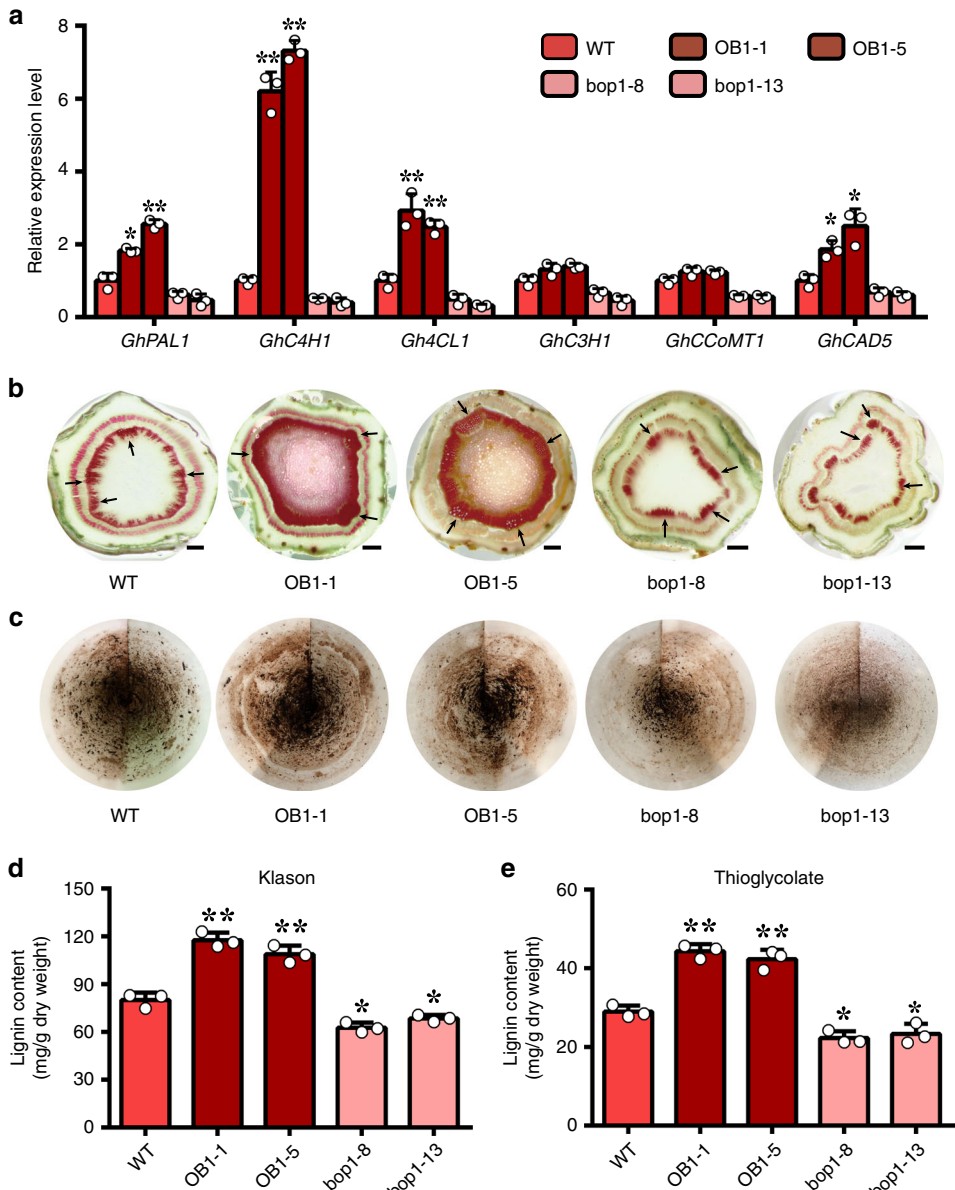

**Fig. 7** GhBOP1 positively regulates lignin deposition. **a** The expression levels of lignin synthesis-related genes in WT and transgenic cotton. Total RNA was extracted from the root of WT and transgenic cotton seedlings. Error bars represent the SD of three biological replicates. The asterisks indicate statistically significant differences compared to corresponding gene expression levels of WT plants, as performed by the Student's *t*-test (*$P < 0.05$, **$P < 0.01$). **b** Phloroglucinol-HCl staining of stem sections at the same position in WT and transgenic cottons. The black arrows indicate vascular bunldes in stem across sections. Scale bars = 0.2 mm. **c** Acid-insoluble lignin residues remained on quantitative filter paper after the Klason extraction. **d**, **e** The lignin content of WT and transgenic cotton was measured by the Klason method (**d**) and thioglycolate (**e**) analysis, respectively. Error bars represent the SD ($n = 18$) of three biological replicates. Asterisks indicate statistically significant differences in comparison to WT, as determined by the Student's *t*-test (*$P < 0.05$, **$P < 0.01$)

decreased in the *GhBP1*-silenced plants and showed markedly higher *GhBOP1* expression levels in stems, reaching 4.3-fold higher levels than in the control plants (Fig. 8b and c). When *GhBP1* was silenced in GhBOP1pro:GUS plants, the GUS staining area almost expanded to the stem and petiole around the LOBs, while GUS staining in the control was retained in the LOBs (Fig. 8d). Thereby, the results of the GUS analysis showed that *GhBP1* knockdown could trigger ectopic expression of GhBOP1, similar to the results obtained for *V. dahliae* infection.

To assess the relationship of the GhBOP1 defence function and the expanding expression by *GhBP1* repression, the *GhBP1*-silenced plants were challenged with *V. dahliae*. These gene-knockdown plants showed higher resistance with a lower rate of diseased plants, fungal biomass and frequency of fungal recovery compared with the control, as shown in Fig. 8e–h, in line with the disease symptoms of the *GhBOP1*-overexpressing plants. Taken together, those results show that GhBP1 and GhBOP1 antagonistically regulate plant defence, possibly due to the expanded GhBOP1 expression beyond the LOBs, which was strictly restricted by GhBP1.

## Discussion
BOP1 and BOP2 redundantly regulate the plant architecture, including development of the leaf, internode, inflorescence, nodule identity, and abscission zone[13,14,16,20,21,43–49]. Notably,

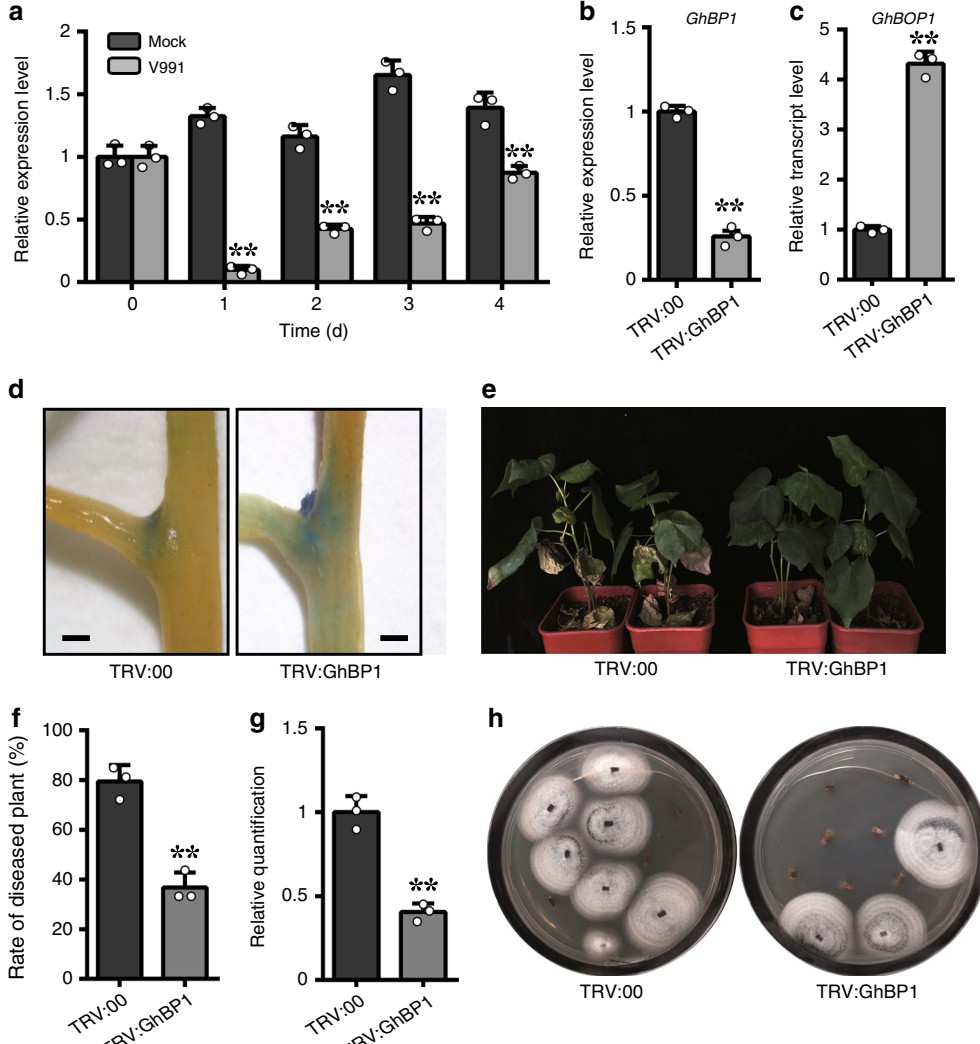

**Fig. 8** *GhBP1* repression of the expanded *GhBOP1* expression beyond the lateral-organ boundary. **a** *GhBP1* expression patterns in cotton roots at 0, 1, 2, 3, and 4 day post-inoculated with *V. dahliae*. *GhUB7* served as an internal control. Error bars represent the SD of three biological replicates. Asterisks indicate statistically significant differences compared to corresponding expression level of mock treated plants, as performed by the Student's *t*-test (*P < 0.05, **P < 0.01). **b** *GhBP1* relative expression levels in roots of TRV:00 and TRV:GhBP1 plants. Error bars represent the SD of three biological replicates. Asterisks indicate statistically significant differences, as determined by the Student's *t*-test (**P < 0.01). **c** *GhBOP1* relative expression levels in roots of TRV:00 and TRV:GhBP1 plants. Error bars represent the SD of three biological replicates. Asterisks indicate statistically significant differences, as determined by the Student's *t*-test (**P < 0.01). **d** GUS staining analysis of the GhBOP1pro:GUS transgenic stems with TRV:00 or TRV:GhBP1 treatment. **e** The disease symptoms of TRV:00 and TRV: GhBP1 plants inoculated with the *V. dahliae*. The photographs were taken at 18 day after *V. dahliae* inoculation. **f** The diseased rate of TRV:00 and TRV: GhBP1 cotton plants at 18 day after inoculated with the *V. dahliae*. Error bars represent the SD of three biological replicates. Asterisks indicate statistically significant differences compared to TRV:00, as determined by Student's *t*-test (**P < 0.01). **g** The relative quantification of the fungal biomass in the infected stems. Error bars represent the SD of three biological replicates. Asterisks indicate statistically significant differences compared to TRV:00, as determined by Student's *t*-test (**P < 0.01). **h** Fungal recovery assay of TRV:00 and TRV:GhBP1 plants at 18 day after *V. dahliae* inoculation. The stem segments at the same position of the plants were placed on PDA medium, and taken photographs at 4 day after culture

BOP1 and BOP2 proteins are extremely similar to NPR1 in domains and structure, a major component in the plant defence pathway, especially in the SA signalling pathway[50]. Thus, it is worthy to investigate whether BOPs participate in plant resistance against the pathogens. In the present study, we found that cotton BOP1 could autonomously expand its expression out of LOBs when the plants were infected by *V. dahliae*, which may be regarded as spatially induced expression. The spatially induced expression of *GhBOP1* could confer economic increases plant resistance to fungi by promoting TGA3 transcriptional activity and lignin accumulation.

BOP1 expression in cotton is specific to LOBs, mainly focusing on the abscission zone between branches and petioles attached to the stem, potentially playing important roles in plant development. When the plants were challenged with *V. dahlia*, the GhBOP1 gene showed autonomously expanded expression out of LOBs, increasing plant resistance to this fungus. Thereby, GhBOP1 in normal plants may regulate development; while the plants are infected by the pathogens, GhBOP1 shows spatially induced expression to confer increases in plant defence. Compatible with this result, *GhBOP1*-overexpressing plants also showed greater resistance to *V. dahliae*. In Arabidopsis, ectopic

expression of BOP1/2 in the stem or pedicle from LOBs functions in the plant architecture[14]. These data indicate that the spatially induced expression of BOPs beyond the LOBs can confer some novel functions including plant defence functions, which is considered an economic model in which only one protein plays roles that are transmit from development to defence according to environmental stresses.

In the present study, we found that GhBOP1 could interact with GhTGA3 and then promote transcriptional levels of downstream disease resistance genes carrying a TGACG element in their promoters. GhTGA3 affected the distribution of GhBOP1 in the cytoplasm and nucleus and GhBOP1 could promote GhTGA3 transcriptional activity, together regulating plant resistance to *V. dahliae*. Additionally, SA defence-related genes showed upregulated and downregulated expression in GhBOP1 overexpression and RNAi plants, respectively, indicating that the function of GhBOP1-GhTGA3 in plant defence involved the SA signalling pathway. In Arabidopsis, researches have indicated that GhTGAs interacting with NPR1 participate in plant defence mainly through the SA signalling pathway[35,36]. Thereby, GhBOP1-GhTGA3 could have same model of action in cotton plant defence as Arabidopsis NPR1-TGAs.

In *GhBOP1*-overexpressing plants, the expression level of lignin synthesis-related genes was upregulated in the stem. The amounts of lignin deposited in the vasculature of *GhBOP1*-overexpressing and *GhBP1*-silenced stems. In the plants treated with *V. dahliae*, the expression patterns of *GhBP1* and *GhBOP1* showed antagonistic trends, in agreement with the results in Arabidopsis. Khan et al.[13] showed that the expression levels of lignin synthesis-relative genes in *bp1* mutant and BOP1/2-overexpressing plants were higher than in WT plants, resulting in greater lignin accumulation in the vascular tissue of the stem. These results suggest that plant BP1 and BOPs can antagonistically regulate lignin synthesis, and BP1 is possibly an upstream regulator of BOPs, which has been proposed in some reports[13,14]. It is regrettable that we did not observe the direct interaction of GhBOP1 with GhBP1. Cotton lignin content is positively related to plant resistance to *V. dahliae*, which has been confirmed in many recent studies[22,24–29]. Thereby, GhBOP1 positively regulates cotton plant defence, partial by the promotion of lignin synthesis.

GhBOP1 mainly participates in cotton plant defence against *V. dahliae* infection through induced expression, VIGS analysis, confirmation of gain- and loss-of-function plants. While *GhBOP2* expression mildly increased in plants infected by *V. dahliae*, and *GhBOP2*-silenced plants showed similar susceptibility to the control. In Arabidopsis, single *bop1* and *bop2* mutants showed the same disease resistance as the wild-type, but *bop1bop2* mutant showed significant susceptibility to *Pseudomonas syringe* (Canet et al., 2012). Therefore, GhBOP1 and GhBOP2 do not show obvious functional redundant in cotton plant disease resistance, which is distinguished from Arabidopsis.

Disease resistance function of NPR1 in Arabidopsis is documented by different *npr1* mutants including *npr1-1* and *npr1-3*[36,51–53]. The NPR1 in SA perception enters the nucleus and promotes TGAs transcriptional activity, resulting in defence in the attack sites and system acquired resistance in other organs uninfected by pathogens[6]. At the same time, NPR1 has been reported to be involved in the resistance-inducing ability of JA[2]. This JA-inducing resistance is considered to come from a cytosolic function of NPR1, likely crucial in cross-talk between SA and JA signalling[36,51–54]. While Canet et al.[17] reported that NPR1 disrupted interaction of BOPs and TGAs, likely attenuating JA-inducing resistance. GhBOP1, as a homolog of NPR1, takes part in cotton plant resistance against *V. dahliae* possibly associated in SA and JA signalling pathways. Addtionally, in

overexpression and RNAi plants, JA-dependent genes, *GhPR3* and *GhPDF1.2*, showed different expression from the wild-type, indicating that the role of GhBOP1 in disease resistance is associated with JA signalling pathway.

Distribution of NPR1 between cytoplasm and nucleus is crucial to confer plant disease resistance, since NPR1 in SA perception becomes an active monomer, which enters into the nucleus and interacts with TGA transcriptional factor. It has been proposed that BOPs locate in both cytoplasm and nucleus in multiple species like NPR1 distribution in the cell[16,21,55]. In this study, GhBOP1 was also distributed in cytoplasm and nucleus in plants and *V. dahliae*-inoculated plants. And GhBOP1 distribution or accumulation in nucleus depended on the presence of GhTGA3. Additionally, since GhBOP1 seems to lack a canonical nuclear localization sequence unlike NPR1 protein and is larger protein in size than the protein for passive diffusion of proteins through nuclear pores[56], we speculate that there is a potential mechanism for GhBOP1 to shuttle between cytoplasm and nucleus. Collectively, GhBOP1 in disease resistance is a similar manner to that of NPR1, and in response to pathogen infection, may accumulate in the nucleus and/or become activated so that they may complex with TGA3 to promote the transcription of defence-related genes.

In plants, BOP expression is limited to the LOB, regulating developmental process[14]. Here we found that *GhBOP1* expression could be spatially induced out of LOBs by *V. dahliae* infection, resulting in increased plant defence. GhBOP1 interaction with GhTGA3 promoted its activation activity, which increased the transcriptional level of downstream resistant-related genes. Additionally, the expanded GhBOP1 expression out of LOBs could upregulate the expression of lignin biosynthesis genes coupled with GhBP1 repression during *V. dahliae* infection, promoting lignin accumulation to confer increasing plant resistance to this fungus. Figure 9a diagrammatically shows that GhBOP1 expression was normally retained in LOBs, functioning in plant development, while GhBOP1 autonomously expanded out of LOBs when plants encountered pathogen attack. Overall, GhBOP1 is an economically positive regulator that participates in plant defence possibly due to the GhBOP-GhTGA3 complex, similar to the NPR1-TGA module in Arabidopsis, and lignin accumulation by co-regulation with GhBP1.

## Materials and methods

**Plant materials, growth conditions, and transformation**. The seeds of *G. hirsutum* cv. CCR35 cotton plants were used in this article and reaped by self-pollination. The cotton plants were grown in soil (roseite: organic matter soil = 1:3) or Hoagland nutrient solution under an illumination incubator with an 8-h/16-h dark/light photoperiod at 25 °C and a relative humidity of 60%.

The tobacco (*N. benthamiana*) was cultivated in soil in the illumination incubator with an 8-h/16-h dark/light photoperiod at 25 °C and a relative humidity of 60%.

The CDS sequence of *GhBOP1* was cloned into the pBIN438 vector to generate *GhBOP1* overexpression vector, named pBIN438-GhBOP1. The specific sequence of *GhBOP1* was inserted into RNAi vector pHANNIBAL to generate the *GhBOP1*-RNAi vector. The recombinant vectors were used to transform into *G. hirsutum* cv. CCR35 by *A. tumefaciens* (LBA4404)-mediated transformation according to previous methods[57], respectively. The primers used for transformation were listed in Table 1.

**Phylogenetic analysis**. The neighbour-joining method in MEGA programme version 5.0 was used to generate the phylogenetic tree of the BOPs[58]. Sequence alignment was performed using DNAMAN 7.0 software for BOP homologue analysis.

**RNA and RT/qRT-PCR reaction**. Total RNA of cotton or tobacco plants was extracted using the Plant Total RNA Extraction Kit (Sangon Biotech, Shanghai, China). For first-strand cDNA synthesis, 1 μg of total RNA was carried out the reverse-transcription reaction using TransScript First-Strand cDNA Synthesis kit (TransGen). All cDNA samples were diluted by 20-fold and stored at −70 °C until further analysis. RT/qRT-PCR was employed to monitor related genes expression level in plant tissues and treatment samples. For qRT-PCR analysis, the SYBR

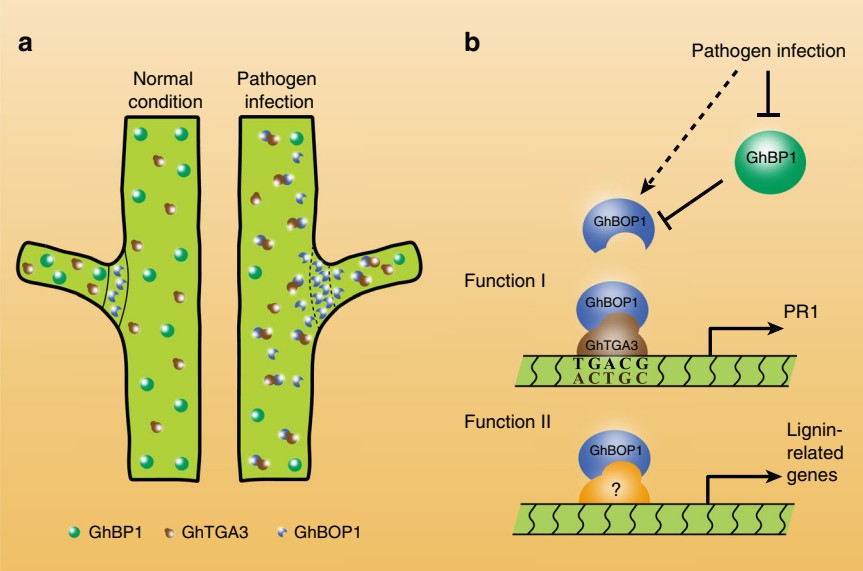

**Fig. 9** *GhBOP1* autonomously expanding expression beyond LOBS increases cotton plant defence against *V. dahliae*. **a** Schematic diagram showing GhBOP1 can expand expression out of LOBs in cotton stem after pathogen infection. The area of solid or point lines surrounding represent LOBs. The green, brown, and blue spheres represent GhBP1, GhTGA3, and GhBOP1, respectively. **b** Working model of GhBOP1 in plant defence process. GhBP1 expression is repressed under the fungus infection, resulting in *GhBOP1* ectopic expression. GhBOP1 acts as a positive regulator participating in plant defence through two functions. Function I, GhBOP1 interacts GhTGA3 and promotes GhTGA3 activation activity for *GhPR1* promoter. Function II, GhBOP1 interacts with an unknown protein, which could be binding the *cis*-element of lignin synthesis-related genes, enhancing lignin accumulation

Green Real-Time PCR Master Mix (TsingKe, Beijing, China) was used on the CFX96 TouchTM Real-Time PCR detection systems (Bio-Rad, Foster City, United States). The *Ntactin* and *GhUb-7* gene were used as internal control for normalization, respectively. All reactions were performed in three biological repeat with three technical repeats. The primers used are shown in Table 1.

**DNA extraction and Southern blotting**. The cotton genomic DNA was extracted from leaves by Plant Genome Extraction Kit (TianGen, Beijing, China) according to the instruction manual. To determine the T-DNA inserted copy numbers of *GhBOP1* transgenic plants, a southern blot analysis was performed according to a previously described method[59]. Briefly, a total of 20 μg genomic DNA was digested by *Hin*d III for 24 h, then separated by electrophoresis on a 0.8% agarose gel. After that, transfer the separated DNA to the nylon membrane (Hybond-N+; Amersham, Buckinghamshire, UK). The amplicon of *NPTII* fragment was used as a probe, which was labeled with α-[$^{32}$P] through a random primer labeling kit (Promega, Madison, WI, United States).

**Pathogen infection and disease assay**. *V. dahliae* strain V991, a highly aggressive defoliating isolate, was stored in 20% glycerol at −80 °C. For activation, 10 μL conidia in glycerol suspension were cultured on potato dextrose agar plates for 3 day at 25 °C in the dark, and then the activity of the fungus was transferred into Czapek's medium for culturing under 200 rpm and 25 °C. Three days later, we collected the fungal conidia and adjusted the concentration to $10^5$ conidia/mL with sterile distilled water containing 5% sucrose. The roots of 3-week-old plants were washed with tap-water, and then the clean roots were soaked in the *V. dahliae* conidia buffer for 1 min[24]. The treated plants were planted in soil and incubated in a chamber.

To investigate plant defence in response to *V. dahliae* infection, the disease grade was surveyed according to a previous report[60]. The fungal recovery assay and fungal biomass quantification were employed at 18 day post-inoculation according to a previously described method[61,62].

**Histochemical assay of GUS activity**. The histochemical assay of GUS activity was performed according to a previously described method[63]. In brief, detached fresh tissues were immediately incubated in 95% acetone for 3 h at 4 °C. The incubated tissues were washed two times with 100 mM sodium phosphate buffer (pH 7.0) and then transferred into staining solution [50 mM PBS, 2 mM K$_4$Fe (CN)$_6$, 2 mM K$_3$Fe(CN)$_6$, 10 mM EDTA, 0.1% Triton-100 and 1 mg/mL 5-bromo-4-chloro-3-indolylglucuronide] for 18–36 h at 37 °C. Next, the stained tissues were washed with buffer containing a gradually increasing content of ethanol from 35 to 95% for 10 min, and photographed with a stereomicroscope (DM2500; Leica, Germany). For quantitative analysis of GUS activity, proteins were extracted in lysis buffer (100 mM PBS, 10 mM EDTA, 0.1% Triton-100, 0.1% SDS and 10 mM

β-mercaptoethanol). The 2 mM 4-MUG was added to the reaction buffer to start the GUS reaction, and finally, 200 mM Na$_2$CO$_3$ was added to stop the reaction at 0, 10, 20, 30, and 40 min, respectively. The fluorescence of 4-MU was measured at an excitation wavelength of 365 nm and emission wavelength of 455 nm using the Infinite 200 PRO multimode reader (Tecan, Switzerland).

**VIGS mediated by Agrobacterium**. Cotton VIGS was conducted according to previously described methods[64]. In brief, the low similarity 5'UTR sequences of *GhBOP1* and *GhBOP2* were used as target to silence *GhBOP1* and *GhBOP2* by VIGS (Supplementary Fig. 11). The specific fragments of the *GhBOP1*, *GhBOP2*, *GhTGA3*, and *GhBP1* genes were amplified by PCR and cloned into the tobacco rattle virus (TRV) vector pTRV2, named pTRV2-GhBOP1, pTRV2-GhBOP2, pTRV2-GhTGA3, and pTRV2-GhBP1, respectively. Primers used in VIGS were listed in Table 1. The pTRV2-gene and assistant vector pTRV1 were transformed into the *A. tumefaciens* GV3101 strains by electroporation, respectively. Agrobacteria containing different vectors were cultured in LB medium at 28 °C for 2 d, collected and resuspended in MMA solution. The suspension concentration was adjusted to an OD600 value of 1.2. For VIGS infiltration, each Agrobacterium containing the pTRV2-gene was equally mixed with the assistant vector pTRV1 strain, respectively. Then, the mixed cultures were incubated at 25 °C for ~3 h in the dark and then injected into cotyledons of 6-day-old cotton seedlings through a 1-mL needleless syringe.

**Yeast two-hybrid assay**. For the directed yeast two-hybrid assays, the Matchmaker Gold System was employed according to the manufacturer's instructions (Clontech, Mountain View, CA). The coding region of *GhBOP1* was cloned into the BD vector pGBKT7, named BD-GhBOP1. The coding regions of the *GhTGA1/2/3/4/5* genes were cloned into the AD vector pGADT7 to produce AD-GhTGA1/2/3/4/5, respectively. The recombinant plasmid BD-GhBOP1 and each AD-gene construct were co-transformed into the yeast strain AH109.

**Transient expression in *N. benthamiana***. To activate *A. tumefaciens*, each strain was grown in LB medium at 28 °C overnight. The activated strains were then cultured in 20 mL LB liquid medium with 50 μg/mL kanamycin and 40 μg/mL rifampicin at 200 rpm and 28 °C. The cultures were centrifuged, and then resuspended in MMA buffer (10 mM MgCl$_2$, 10 mM MES and 200 μM acetosyringone), and finally adjusted the OD600 value to 1.2. Subsequently, the resuspended cultures were incubated at room temperature for 3 h. For the co-expression analysis, two or three strains carrying appropriate reconstructs were equally mixed and infiltrated into *N. benthamiana* young leaves using a 1-mL needleless syringe. Leaf samples were collected at 36 h after agro-infiltration to analyse the GUS activity and gene expression. Agro-infiltration was carried out at least three times for the same experiment.

### Table 1 Primers used in this study

| Gene name | Purpose | Forward primer | Reverse primer |
|---|---|---|---|
| GhBOP1 | pHANNIBAL-1; RNAi | CGGGAATTCCAGAAACAAACCACAGCACTCC | CGGGGTACCTAGGGTTGTCTTTCACGAGTAAA |
| GhBOP1 | pHANNIBAL-2; RNAi | CGGTCTAGACAGAAACAAACCACAGCACTCC | CGCGGATCCTAGGGTTGTCTTTCACGAGTAAA |
| GhBOP1 | pBin438; OE | CGCGGATCCATGAGTAGCTTTGAGGAGTCCTTAAG | ACGCGTCGACCTAGAAATCATGAGAGTGATGGTACATT |
| ProGhBOP1 | pBIN121 | CCCAAGCTTTAGGCTTCACTTTGGCTTCGAT | CGCGGATCCTTTAGGGTTGTTTTTCACGAGAAAAAG |
| GhBOP1 | pGBKT7 | CGGGAATTCATGAGTAGCTTTGAGGAGTCCTTAAG | CGCGGATCCCTAGAAATCATGAGAGTGATGGTACATT |
| GhTGA1 | pGADT7 | CGGGAATTCATGGGCCAACACTTGG | CGCGGATCCCTAAGGCAAGGAAGGTTCAGGA |
| GhTGA2 | pGADT7 | CGGGAATTCATGGGCAGTAGAACGTTGAAAAAT | CGCGGATCCTCACTCGCGAGGTCTAGCCAG |
| GhTGA3 | pGADT7 | CGGGAATTCATGACAATATACGAGCAACTAAACC | CGCGGATCCTCAGAGTGAACTGAGAGCTCGG |
| GhTGA4 | pGADT7 | CGGGAATTCATGGCTAATGGGCTCATACTCCT | CGCGGATCCTTATGCAGGTTCACGAGCACAAG |
| GhTGA5 | pGADT7 | CGGGAATTCATGCCGGGTTTTGACTCACA | CGCGGATCCTCACTCTCGGGGTCGGGC |
| GhBOP1 | pCambia-NLuc | CGCGGATCCATGAGTAGCTTTGAGGAGTCCTTAAG | ACGCGTCGACCTAGAAATCATGAGAGTGATGGTACATT |
| GhTGA1 | pCambia-CLuc | CGGGGTACCATGGGCGACCCAACACTTGG | ACGCGTCGACCTAAGGCAAGGAAGGTTCAGGA |
| GhTGA2 | pCambia-CLuc | CGGGGTACCATGGGCAGTAGAACGTTGAAAAAT | ACGCGTCGACTCACTCGCGAGGTCTAGCCAG |
| GhTGA3 | pCambia-CLuc | CGGGGTACCATGACAATATACGAGCAACTAAACC | ACGCGTCGACTCAGAGTGAACTGAGAGCTCGG |
| GhTGA4 | pCambia-CLuc | CGGGGTACCATGGCTAATGGGCTCATACTCCT | ACGCGTCGACTTATGCAGGTTCACGAGCACAAG |
| GhTGA5 | pCambia-CLuc | CGGGGTACCATGCCGGGTTTTGACTCACA | ACGCGTCGACTCACTCTCGGGGTCGGGC |
| GhBOP1 | pPZP111; gene-GFP | CGCGGATCCATGAGTAGCTTTGAGGAGTCCTTAAG | GTGGCCGCGGTAGAAATCATGAGAGTGATGGTACATT |
| GhTGA3 | pPZP111; gene-GFP | CGCGGATCCATGACAATATACGAGCAACTAAACC | GTGGCCGCGGCAGAGTGAACTGAGAGCTCGG |
| GhBOP1 | pMAL-p2X; MBP-GhBOP1 | CGCGGATCCATGAGTAGCTTTGAGGAGTCCTTAAG | CCCAAGCTTCTAGAAATCATGAGAGTGATGGTACATT |
| GhTGA3 | pGEX6P-1; GST-GhTGA3 | CGCGGATCCATGACAATATACGAGCAACTAAACC | CGCGAATTCTCAGAGTGAACTGAGAGCTCGG |
| ProGhPR1 | pBIN121 | CCCAAGCTTTCACATAGATTTGGTGGGTAGGG | CGCGGATCCTTTGAAGTTTGATTCTATAAGAATATTGC |
| GhTGA3 | pHANNIBAL-1; RNAi | CGGGAATTCCTGAGCGTTTCTTCCATTGGAT | CGGGGTACCGCTCGGAGACGGTTGAAGTAA |
| GhTGA3 | pHANNIBAL-2; RNAi | CGGTCTAGACTGAGCGTTTCTTCCATTGGAT | CGCGGATCCGCTCGGAGACGGTTGAAGTAA |
| GhBOP1 | pTL156; VIGS | CGGGGTACCCAGAAACAAACCACAGCACTCC | CGCGGATCCTAGGGTTGTCTTTCACGAGTAAA |
| GhBOP2 | pTL156; VIGS | CGGGGTACCCTAAAGTTTTCTCTTTTCTCTGTCTATC | CGCGGATCCGGTTGTTTTCCCAGAGGAATAC |
| GhBP1 | pTL156; VIGS | CGGGGTACCCGGGAAGAACTAACGAGGCC | CGCGGATCCGTTGATTCAGCCAATGCCACC |
| GhTGA3 | pTL156; VIGS | CGGGGTACCTGAGCGTTTCTTCCATTGGAT | CGCGGATCCGCTCGGAGACGGTTGAAGTAA |
| NtHistone 3 | RT-PCR | TCCTGGGCAATTTCACGAACAAGC | TGCCCGTAAATCTACTGGAGGCAA |
| GhUB-7 | qPCR | GAAGGCATTCCACCTGACCAAC | CTTGACCTTCTTCTTCTTGTGCTTG |
| GhBOP1 | qPCR | AACCCACCAACTTCAACTGCGA | TCACCCTCCATTCTCGAACCCA |
| GhBOP2 | qPCR | AGAGCTTGGTGCAGCTGATGTT | TGGTGGTCTAAAAGCACCGCAA |
| ITS | qPCR | AAAGTTTTAATGGTTCGCTAAGA | CTTGGTCATTTAGAGGAAGTAA |
| GhPAL1 | qPCR | TCCAGGACAAATTGAGGCAGCG | CCAAGCCACTGTGGAGAAGTCC |
| GhC4H1 | qPCR | CCGAACCCGACCATCTAAAGC | GCAGGGATGTCATACCCACCAAG |
| Gh4CL1 | qPCR | GTGTCTTGCCTTTATTCCACATTTAC | TTCTTAGCCAACAACACCACCAAC |
| GhC3H1 | qPCR | ACTATTATTGGACTTCTTTGGGACAT | ATCAGTTTCAGACATCACCCTTTC |
| GhCCoMT1 | qPCR | CACCTGGGTCACAATCCCTTAC | CCAACTGTGGCACGGCAAT |
| GhCAD5 | qPCR | CCGACCATGCGACTCAGACAAT | CTTGGGTGGGTTTTCCGTCAGT |
| GhPR1 | qPCR | GCTCTTGTAGGTGCTCTTGTTCTTCCCT | CTGGTTGTGAACCCTTAGATAATCTTGTGG |
| GhPR2 | qPCR | CAATCTCCCTTGCTCGTGAATCTCTACC | CGTTATCAACAGTGGACTGGGCGG |
| GhPR3 | qPCR | TTAACGGCCTCCTCGAAGCTGCTATTT | CGCAACATAAACAGTGAAACATCATTGGAA |
| GhPDF1.2 | qPCR | CAAGTGGGACATGGTCAGGGGTT | CACTTGTGTGCTGGGAAGACATAGTTGC |
| GhBP1 | qPCR | AGCGGCTCCAACGTGAGTAAT | GCTTCCAAGAGGTTAGAGTACTGAGG |

Solid lines under sequences indicate restriction enzyme cutting sites

**Immunoblot analysis**. The roots of wild-type and *GhTGA3*-silenced cotton plants challenged with *V. dahliae* or mock were ground into fine powders in liquid nitrogen. The proteins of nucleus and cytoplasm were extracted using Plant Nuclear and Cytoplasmic Protein Extraction Kit (Bestbio, Shanghai, China). Both nuclear and cytoplasmic proteins were quantified by Bradford assay, and 40 µg of the proteins was subjected to SDS-PAGE. The immunoblot experiment was performed using the antibodies (1:2000 dilution) raised against GhBOP1, β-Actin (1:5000 dilution; EarthOx, San Francisco, CA) and Histone 3 (1:5000 dilution; EarthOx) as the primary antibodies, and the horseradish peroxidase conjugated goat anti-rabbit/mouse IgG (1:3000 dilution; Sungene Biotechnology, Tian Jin, China) as the secondary antibody. The uncropped blots were presented in Supplementary Fig. 12.

**Electrophoretic mobility shift assay**. The coding region of GhBOP1 was inserted into the pMAL-p2X vector to generate a fusion protein: maltose-binding protein (MBP)-GhBOP1. The coding region of GhTGA3 was cloned into the pGEX6P-1 vector to produce a fusion protein: glutathione S-transferase (GST)-GhTGA3. The recombinant proteins were expressed in *E. coli* strain BL21 and purified using corresponding affinity columns. For the EMSA assay, the biotin-labelled probes and a Pierce LightShift Chemiluminescent EMSA kit (Thermo, Rockford, IL) were used. The binding reaction was performed in a 20 µL reaction mixture at room temperature for 30 min and then separated on a native 6% polyacrylamide gel in 0.5 × Tris-borate/EDTA buffer. The labelled probes were detected based on the manufacturer's instructions. The uncropped blots were presented in Supplementary Fig. 12.

**Dual-luciferase reporter assay**. The coding region of GhTGA3 was cloned into the expression vector pRT-BD to construct the BD-GhTGA3 effecter plasmid. The 5 × Gal4-Luc and Renilla Luc gene were used as the reporter and internal control,

respectively. The dual-luciferase reporter assay was conducted according to the method described by Ohta et al.[38]. The samples of co-transformed Arabidopsis protoplasts were cultured in the dark for 12 h at 25 °C, as described by He et al.[39]. The Luc assay was carried out using the Promega dual-luciferase reporter assay system and measured using a GloMax 20/20 luminometer (Bio-Rad).

**Firefly luciferase complementation imaging assay**. The luciferase complementation imaging assay is usually employed to validate protein interactions. In this study, the luciferase complementation imaging assay was conducted as described previously[23]. In brief, the coding sequences of *GhBOP1* cloned into the pCAMBIA-NLuc vector to generate NLuc-GhBOP1 and the coding sequences of *GhTAG1/2/3/4/5* were inserted into the pCAMBIA-CLuc vector to produce GhTAGs-CLuc, individually. Agrobacterium carrying the constructs was co-infiltrated into the leaves of *N. benthamiana*. Two days later, the treated leaves were detached and infiltrated with 1 mM luciferin (Promega) and placed in the dark for 10 min. Next, we started to capture the Luc signal and measured the relative Luc activity using a low-light cooled charge-coupled device camera (Night owl LB985, Berthold Technologies, Germany).

**Histochemical staining and total lignin content analysis**. Histochemical staining was used to visualize lignin deposition using Wiesner reagent according to Xu et al.[24]. Cotton stems of transgenic plants and WT were crosscut into sections by hand. These stem sections were dipped in phloroglucinol solution (2% in 95% ethanol) for 10 min and then transferred into 18% HCl for 20 min and photographed using a stereomicroscope (DM2500; Leica, Germany). The total lignin content of the stem was determined using the Klason and thioglycolate methods as described previously[65]. The same experiment was conducted with three biological replicates.

**Statistical analysis.** Statistical analyses were performed by one-way ANOVA with Duncan's post-hoc test or two-way ANOVA with student's *t*-test and *P* < 0.05 was considered as significant.

**Accession numbers.** Arabidopsis genes sequence invoved in this work can be found in the TAIR (The Arabidopsis Information Resource, http://www.arabidopsis.org/), accession numbers as following: *AtBOP1* (AT3G57130), *AtBOP2* (AT2G41370), *AtNPR1* (AT1G64280), *AtNPR2* (AT4G26120), *AtNPR3* (AT5G45110), *AtNPR4* (AT4G19660).

Cotton genes sequnce informations obtained from the CottonGen database (https://www.cottongen.org/), accession numbers as following: *GhBOP1* (Gh_A09G1115), *GhBOP2* (Gh_A01G1644), *GhTGA1* (Gh_D11G2225), *GhTGA2* (Gh_A11G2749), *GhTGA3* (Gh_D02G0824), *GhTGA4* (Gh_D10G0995), *GhTGA5* (Gh_A10G1678), *GhBP1* (Gh_A05G1857), *GhPR1* (Gh_D09G0971).

The other species genes sequence were download from NCBI database (https://www.ncbi.nlm.nih.gov/), accession numbers as following: *PtrBPL2* (XP_002308905.1), *PtrBPL1* (XP_002323261), *NtBOP4* (AFK30390.1), *LjBOP2* (AEM62768.1), *GmNPR5* (XP_003521136.1), *PsBTB/POZ* (AET34790.1), *MtBTB/POZ* (AET34786.1), *HvCul4* (BAJ91942.1).

**Reporting summary.** Further information on research design is available in the Nature Research Reporting Summary linked to this article.

## Data availability
The authors declare that all data supporting the findings of this study are available within the paper and its supplementary information. Source data underlying the graphs presented in the main figures is available in Supplementary Data 1.

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

## Acknowledgements

This study was supported by grants from the National Natural Science Foundation of China (31771848 and 31471544), the State Key Laboratory of Cotton Biology Open Fund (CB2017B04).

## Author contributions

J.W. and G.X. conceived the study and designed the experiments. Z.Z., P.W., X.L., C.Y., Y.T., and Z.W. performed the experiments. G.H., X.G., and J.W. analyzed the data. Z.Z. and J.W. wrote the manuscript. All of the authors read and approved the final manuscript.

## Additional information

**Competing interests:** The authors declare no competing interests.

