## [Peer Review File · Communications Biology]

Reviewers' comments:

Reviewer #1 (Remarks to the Author):

The manuscript entitled “GhBOP1 actively autonomously expanding expression out of lateral-organ boundaries increases cotton plant defense against *Verticillium dahliae*” by Zhang et al reveals an important potential mechanism of cotton plant defense. Majorly based on genetic and biochemical analyses, a spatially induced expression gene, GhBOP1, were confirmed to participate in plant resistance against *V. dahliae* through both regulation of GhTGA3 in downstream genes' expression with TGACG cis-elements of promoters and lignin accumulation in vascular section companied by GhBP1 repression. More importantly, this study demonstrated that the GhBOP1, primarily regulating differentiation of lateral-organ boundaries, can be spatially induced by the fungi infection, which suggested that GhBOP1 is an economical regulator to trade off plant development and defense. The findings of this study contribute significantly to plant innate immunity.

This manuscript offers solid experimental data to support corresponding results, and is well organized and easy to understand. However, some minor questions should be considered to improve this manuscript.

1. Throughout the manuscript, some “PR1” should be changed into “GhPR1”; similar typo errors should also be revised, such as “PR2”, “TGA3”.
2. “PR1pro:GUS” and “GhPR1pro:GUS” should be consistent in main text and Figures.
3. In materials and methods, line 644-646: “coding sequences of GhTAG1/2/3/4/5 and GhNPR1 were inserted into the pCAMBIA-CLuc vector to produce GhTAGs-CLuc and GhNPR1-CLuc,” this description of experiments regarding GhNPR1 is not involved in this study. Thereby, please delete them.
4. In Figure 1 D, a scale of bar is shortage, please add it.
5. Figure 1 F, the title of Y-axis is some wrong, please revise it. The same wrong is followed by Figure 6B, 6D
6. In Figure 2B, significance tests about disease grades between transgenic plants and wild-type plants should be performed.
7. The names of samples In Figure 6B is not consistent with those of samples in Figure 6A. Please revise.
8. Figure 7B, the scale of bars is missing, please add them. And in this figure, the various parts of vascular tissue need to be labeled by arrows.

Reviewer #2 (Remarks to the Author):

This manuscript describes NPR1-like, BTB-ankyrin family protein GhBOP1 playing positive role in cotton plants in resistance to *Veticillium Dahliae*, which causes *Verticillium* wilt disease. Previous reports have shown that the expression of BOP genes is generally restrained in lateral-organ boundaries and BOP1/2 were involved in the development of the leaf and architecture. Here the authors showed that the expression of GhBOP1 expand out of lateral-organ boundaries in cotton stem after *V. dahliae* infection.

Further, the authors proved that like in Arabidopsis, GhBOP1 interacts with transcription factor GhTGA3, positively regulates defense related genes expression by enhancing transcription activity of GhTGA3. They also confirmed that GhBOP1 is a positive regulator of lignin synthesis, similar as BOP1/2 in Arabidopsis. And as in Arabidopsis, GhBP1 represses expression of GhBOP1 outside of lateral-organ boundaries. Thus, the main new finding of the manuscript is that spatially induced expression of one protein, GhBOP1, plays roles in development in lateral-organ boundaries and plays roles in disease resistance outside of lateral-organ boundaries. In general, the manuscript was well written in a clearly logical way. The conclusions were sound based on data showed. The work might have less novelty to plant immunity community, but would be interest to others in crop disease resistance field.

One issue about the data is that all the quantification figures, for example qPCR data, LUC activities, should do statistical analysis from at least three biological replicates.

Reviewer #3 (Remarks to the Author):

Major Comment:

1. line 138 Results: what is the difference in CDS sequences between GhBOP1 in At and Dt subgenome, and which is employed in this study for further analysis?
2. Line 141 and 148 results: according to the Figure S1B, GhBOP1 and GhBOP2 show high similarity in amino acid sequence, how to design the VIGS of GhBOP1 and GhBOP2 and distinguish each other?
3. Line 162 results: how about the expression pattern of GhBOP1 after fungus inoculation, especially in roots where the fungus initiate infect plants?
4. Line 69, 83 and 148: in ref6, bop1 and bop2 show functional redundant, and the ability of NPR1 to disrupt BOP interactions with TGAs in Arabidopsis. Give more comments in discussion according to this study that GhBOP1 could activate SA and JA in same time.
5. Line 275 results: how to explain GhBOP1 enter nucleus after challenging with *V. dahliae*?

Minor Comments:

1. line 1 and 34: GhBOP1 in title is better with full name of gene BLADE-ON-PETIOLE 1. Similar in keywords.
2. Line 68: Make sure about the reference. I am not sure that there are statements that bop1bop2 mutant showed no change in resistance to pathogens in ref4 and ref12.
3. Line 75: Ref11, 17 and 18 do not support the statement that bop1 can interact with bZIP TFs. Ref 16 support it.
4. line 144 '*V. dahlia*' should be '*V. dahliae*'. There are more in the manuscript.
5. line 257. '*Nicotianabenthamiana*' should be '*Nicotiana benthamiana*'.
6. Line 532 Materials and Methods: more details of transformation and molecular characterization of transgenic plants need to be provided, like how many independent lines obtained and southern blotting results.
7. Line 588 Materials and Methods: more details of VIGS of GhBOP1 and GhBOP2 need to be provided.

Reviewer #1 (Remarks to the Author):

The manuscript entitled “GhBOP1 actively autonomously expanding expression out of lateral-organ boundaries increases cotton plant defense against *Verticillium dahliae*” by Zhang et al reveals an important potential mechanism of cotton plant defense. Majorly based on genetic and biochemical analyses, a spatially induced expression gene, GhBOP1, were confirmed to participate in plant resistance against *V. dahliae* through both regulation of GhTGA3 in downstream genes’ expression with TGACG cis-elements of promoters and lignin accumulation in vascular section accompanied by GhBP1 repression. More importantly, this study demonstrated that the GhBOP1, primarily regulating differentiation of lateral-organ boundaries, can be spatially induced by the fungi infection, which suggested that GhBOP1 is an economical regulator to trade off plant development and defense. The findings of this study contribute significantly to plant innate immunity.

This manuscript offers solid experimental data to support corresponding results, and is well organized and easy to understand. However, some minor questions should be considered to improve this manuscript.

1. Throughout the manuscript, some “PR1” should be changed into “GhPR1”; similar typo errors should also be revised, such as “PR2”, “TGA3”.

Response: Thanks for your elaborate suggestions; these typo errors mentioned have been revised throughout the manuscript.

2. “PR1pro:GUS” and “GhPR1pro:GUS” should be consistent in main text and Figures.

Response: We have corrected “PR1pro:GUS” into “GhPR1pro:GUS” in the manuscript.

3. In materials and methods, line 644-646: “coding sequences of GhTAG1/2/3/4/5 and GhNPR1 were inserted into the pCAMBIA-CLuc vector to produce GhTAGs-CLuc and GhNPR1-CLuc,” this description of experiments regarding GhNPR1 is not involved in this study. Thereby, please delete them.

Response: Your suggestion is right. We have deleted the description about “GhNPR1” in Materials and Methods.

4. In Figure 1 D, a scale of bar is shortage, please add it.

Response: Thanks for your suggestion, the corresponding scale bar was added into Figure 1D.

5. Figure 1 F, the title of Y-axis is some wrong, please revise it. The same wrong is followed by Figure 6B, 6D

Response: We have corrected the title of Y-axis in Figure 1G and 6B as “Gus activity (nmol-4MU min⁻¹ mg⁻¹ protein)”, and replaced the title of Y-axis in Figure 6D as “Gus activity (pmol-4MU min⁻¹ mg⁻¹ protein)”

6. In Figure 2B, significance tests about disease grades between transgenic plants and wild-type plants should be performed.

Response: We have added significance tests about the data of disease grades between transgenic and wild-type plants shown in Figure 2B.

7. The names of samples In Figure 6B is not consistent with those of samples in Figure 6A. Please revise.

Response: We have modified the sample names shown in Figure 6B to make sure the consistence with those in Figure 6A.

8. Figure 7B, the scale of bars is missing, please add them. And in this figure, the various parts of vascular tissue need to be labeled by arrows.

Response: The corresponding scale bars were added into each photographs of Figure 7B. And the black arrows were added to indicate the vascular tissues.

Reviewer #2 (Remarks to the Author):

This manuscript describes NPR1-like, BTB-ankyrin family protein GhBOP1 playing positive role in cotton plants in resistance to *Veticillium Dahliae*, which causes Verticillium wilt disease. Previous reports have shown that the expression of BOP genes is generally restrained in lateral-organ boundaries and BOP1/2 were involved in the development of the leaf and architecture. Here the authors showed that the expression of GhBOP1 expand out of lateral-organ boundaries in cotton stem after *V. dahliae* infection. Further, the authors proved that like in Arabidopsis, GhBOP1 interacts with transcription factor GhTGA3, positively regulates defense related genes expression by enhancing transcription activity of GhTGA3. They also confirmed that GhBOP1 is a positive regulator of lignin synthesis, similar as BOP1/2 in Arabidopsis. And as in Arabidopsis, GhBP1 represses expression of GhBOP1 outside of lateral-organ boundaries. Thus, the main new finding of the manuscript is that spatially induced

expression of one protein, GhBOP1, plays roles in development in lateral-organ boundaries and plays roles in disease resistance outside of lateral-organ boundaries. In general, the manuscript was well written in a clearly logical way. The conclusions were sound based on data showed. The work might have less novelty to plant immunity community, but would be interest to others in crop disease resistance field.

Response: Thank the reviewer for her/his positive comments to our study.

One issue about the data is that all the quantification figures, for example qPCR data, LUC activities, should do statistical analysis from at least three biological replicates.

Response: Thank you for your professional suggestion about the missing of statistical analysis of quantification data shown in all figures. This opinion is valuable and very helpful for revising and improving our study. We have performed the statistical analysis for data from each quantification figures including qPCR data and LUC activities.

We have made corrections which we hope meet with approval.

Reviewer #3 (Remarks to the Author):

Major Comment:

1. line 138 Results: what is the difference in CDS sequences between GhBOP1 in At and Dt subgenome, and which is employed in this study for further analysis?

Response: A 99.6% similarity of coding sequences between At- and Dt-subgenome of *GhBOP1* is shared, only 6 out of 1449 nucleotides exhibiting difference. And in amino acid sequence, there are just 3 mismatched amino acids, but the mismatched amino acids show similar property. Thus, the CDS of *GhBOP1* located in At subgenome (access number: Gh_A09G1115) was cloned to use in the present research. We supplemented the information in line 140 at Results and shown in Figure S2 and S3.

2. Line 141 and 148 results: according to the Figure S1B, GhBOP1 and GhBOP2 show high similarity in amino acid sequence, how to design the VIGS of GhBOP1 and GhBOP2 and distinguish each other?

Response: There is high conservation between GhBOP1 and GhBOP2 in amino acid sequence, but the 5'UTR sequences of the two mRNAs show low similarity. Therefore, the 5'UTR sequences were used to perform VIGS of *GhBOP1* and *GhBOP2*. We have added the information in line 663 at Materials and Methods and shown in Figure S11.

3. Line 162 results: how about the expression pattern of GhBOP1 after fungus inoculation, especially in roots where the fungus initiate infect plants?

Response: We understand the reviewer's meaning. We have supplemented the GUS staining analysis in the roots of GhBOP1pro:GUS transgenic plants after *V. dahliae* inoculation or mock treatment. These results have been added in line 174 at Result section, now it is reading:

"Additionally, the GUS staining color in root after inoculated with *V. dahliae* was darker than that of the mock treatment (Fig S4), indicating the GhBOP1 expression level in roots was induced by the fungi, consistent with the transcription level of *GhBOP1* in inoculated roots as shown in Fig 1B."

4. Line 69, 83 and 148: in ref6, bop1 and bop2 show functional redundant, and the ability of NPR1 to disrupt BOP interactions with TGAs in Arabidopsis. Give more comments in discussion according to this study that GhBOP1 could activate SA and JA in same time.

Response: The reviewer's opinion is constructive in improving this manuscript. Thereby, we add a paragraph to discuss the functional redundant of GhBOP1 and GhBOP2, the disease resistance of GhBOP1 involved in SA and JA signalings including GhBOP1 interaction with GhTGA3, and GhBOP1 distribution in cytoplasm and nucleus. The informations were added in line 256 at the Discussion section.

5. Line 275 results: how to explain GhBOP1 enter nucleus after challenging with V.

dahliae?

Response: GhBOP1 can localize in the cytoplasm and nucleus. And the GhBOP1 contents in cytoplasm and nucleus increased when the wild type plants were inoculated with *V. dahliae*. However, when the *GhTGA3*-silenced plants were challenged with the fungi, GhBOP1 contents in nucleus decreased compared with the wild type. These results indicated that the distribution of GhBOP1 in a cell can depend on the presence of GhTGA3. In the Discussion section, we have added information about BOPs subcellular localization in multiple species and possible shuttling reasons between cytoplasm and nucleus. The corresponding informations were added in line 256 at the Discussion section.

Minor Comments:

1. line 1 and 34: GhBOP1 in title is better with full name of gene BLADE-ON-PETIOLE 1. Similar in keywords.

Response: In title and keywords, “GhBOP1” have been replaced with “BLADE-ON-PETIOLE 1”

2. Line 68: Make sure about the reference. I am not sure that there are statements that bop1bop2 mutant showed no change in resistance to pathogens in ref4 and ref12.

Response: Thanks you for your meticulous reading, References 4 and 12 has been replaced with new Ref 16 at line 69.

3. Line 75: Ref11, 17 and 18 do not support the statement that bop1 can interact with bZIP TFs. Ref 16 support it.

Response: References 11, 17 and 18 have been replaced with new Ref 16 and 17 at line 77

4. line 144 ‘*V. dahlia*’ should be ‘*V. dahliae*’. There are more in the manuscript.

Response: Throughout the manuscript, we have changed the typo error-‘*V. dahlia*’ into ‘*V. dahliae*’.

5. line 257. ‘*Nicotianabenthamiana*’ should be ‘*Nicotiana benthamiana*’.

Response: ‘*Nicotianabenthamiana*’ have been modified as ‘*Nicotiana benthamiana*’

6. Line 532 Materials and Methods: more details of transformation and molecular characterization of transgenic plants need to be provided, like how many independent lines obtained and southern blotting results.

Response: The transgenic plants were generalized using *Agrobacterium tumefaciens*-mediated genetic transformation methods. We obtained 24 and 26 independent transformants of RNAi and overexpression plants, respectively. We have added the experiment of Southern blotting to analyze copies of T-DNA insertion in the four transgenic lines (bop1-8, bop1-13, OB1-1 and OB1-5), which were used in this study. The statements were added in line 183 and 202 at Results section, southern blotting analysis was shown in Figure S6B.

The corresponding methods had been added in line 621 at Materials and Methods section.

7. Line 588 Materials and Methods: more details of VIGS of GhBOP1 and GhBOP2 need to be provided.

We have added more details about VIGS of *GhBOP1* and *GhBOP2* in line 663 at Materials and Methods section.

REVIEWERS' COMMENTS:

Reviewer #2 (Remarks to the Author):

In this revised manuscript, the authors performed the statistical analysis for data from quantification figures and thus improved the quality of the figures.

Reviewer #3 (Remarks to the Author):

Thanks for the authors' efforts, I have no more questions.